# Lipids and ions traverse the membrane by the same physical pathway in the nhTMEM16 scramblase

Tao Jiang[1,2], Kuai Yu[3], H Criss Hartzell[3]*, Emad Tajkhorshid[1,2]*

[1]Department of Biochemistry, Center for Biophysics and Computational Biology, University of Illinois at Urbana-Champaign, Urbana, United States; [2]Beckman Institute for Advanced Science and Technology, University of Illinois at Urbana-Champaign, Urbana, United States; [3]Department of Cell Biology, Emory University School of Medicine, Atlanta, United States

**Abstract** From bacteria to mammals, different phospholipid species are segregated between the inner and outer leaflets of the plasma membrane by ATP-dependent lipid transporters. Disruption of this asymmetry by ATP-independent phospholipid scrambling is important in cellular signaling, but its mechanism remains incompletely understood. Using MD simulations coupled with experimental assays, we show that the surface hydrophilic transmembrane cavity exposed to the lipid bilayer on the fungal scramblase nhTMEM16 serves as the pathway for both lipid translocation and ion conduction across the membrane. $Ca^{2+}$ binding stimulates its open conformation by altering the structure of transmembrane helices that line the cavity. We have identified key amino acids necessary for phospholipid scrambling and validated the idea that ions permeate TMEM16 $Cl^-$ channels via a structurally homologous pathway by showing that mutation of two residues in the pore region of the TMEM16A $Ca^{2+}$-activated $Cl^-$ channel convert it into a robust scramblase.
DOI: https://doi.org/10.7554/eLife.28671.001

*For correspondence:
criss.hartzell@emory.edu (HCH);
emad@life.illinois.edu (ET)

**Competing interests:** The authors declare that no competing interests exist.

## Introduction

Different phospholipid species are typically distributed asymmetrically between the two leaflets of the plasma membrane. Dissipation of this asymmetry in response to the elevation of cytoplasmic $Ca^{2+}$ concentration is a ubiquitous signaling mechanism critical for diverse cellular events including blood coagulation, bone mineralization, and cell-cell interaction (*Fadok et al., 1992*; *Fadok et al., 2001*; *Sahu et al., 2007*; *Bratton and Henson, 2008*; *Sanyal and Menon, 2009*; *Bevers and Williamson, 2016*; *Nagata et al., 2016*; *Pomorski and Menon, 2016*). Phospholipid scrambling is mediated by phospholipid scramblases, which harvest the energy of the phospholipid gradient to drive the nonspecific and bidirectional transport of phospholipids between leaflets. Proteins responsible for $Ca^{2+}$-activated lipid scrambling belong to the anoctamin/TMEM16 superfamily (*Suzuki et al., 2010*, *2013*; *Pedemonte and Galietta, 2014*; *Picollo et al., 2015*; *Brunner et al., 2016*; *Whitlock and Hartzell, 2017*). Among the ten mammalian TMEM16 paralogs, TMEM16C, D, F, G, and J have been identified as $Ca^{2+}$-activated phospholipid scramblases while TMEM16A and TMEM16B are $Ca^{2+}$-activated $Cl^-$ channels (CaCCs). The scramblases and ion channels are thought to share a common architecture and mode of $Ca^{2+}$ activation, but the conduction pathways for ion and lipid transport remain incompletely understood.

The fungal nhTMEM16 (*Brunner et al., 2014*, *2016*) was recently crystalized and has become an important model for structural and functional investigation of TMEM16s. nhTMEM16 is a homodimer with 10 transmembrane helices in each subunit. The protein exhibits an unusual ~10 Å wide hydrophilic cavity on the surface of each subunit. This subunit cavity is formed by transmembrane (TM)

helices TM3-TM7. The helices that border the cavity (TM4 and TM6) are amphipathic, with polar and charged residues facing the cavity. *Brunner et al. (2014)* proposed that the hydrophilic cavity provides a pathway for phospholipid head groups to move between membrane leaflets during scrambling while the fatty acid tails remain in the hydrophobic bilayer. This mechanism is also predicted by molecular dynamics simulations (*Stansfeld et al., 2015*; *Bethel and Grabe, 2016*). Because this surface cavity is hydrophilic and involved in transport, we call it the 'aqueduct'. A highly conserved $Ca^{2+}$-binding site for two $Ca^{2+}$ ions is located near the cytoplasmic end of the aqueduct. Equivalent residues are responsible for $Ca^{2+}$ activation of both TMEM16 phospholipid scramblases and ion channels (*Yu et al., 2012*; *Malvezzi et al., 2013*; *Brunner et al., 2014*; *Tien et al., 2014*).

Although the structure of the nhTMEM16 aqueduct provides insight into the site of catalysis, the absence of phospholipids in the crystal structure leaves unanswered the question of how both lipids and ions are conducted. Here, we use a combined computational and experimental approach to elucidate mechanisms of phospholipid scrambling and the molecular basis of regulation by $Ca^{2+}$. Our atomistic molecular dynamics (MD) simulations reveal that ions permeate via the same structural path as phospholipids. Furthermore, simulation-guided mutational analysis identifies amino acids crucial for TMEM16 scrambling activity and provides insight into evolutionary relationships of the TMEM16 family members.

## Results

### Lipid bilayer deformation

Integral membrane proteins contain hydrophobic segments that are in contact with the acyl chains of the lipid bilayer. From an energetic perspective, the length of the lipid-exposed hydrophobic segments would be expected to approximately match the hydrophobic bilayer thickness. However, in nhTMEM16, the aqueduct that faces the membrane bilayer is strongly hydrophilic with polar and charged residues concentrated in the cytoplasmic half of TM4 and the extracellular half of TM6 (*Figure 1—figure supplement 1*). This staggering of the transmembrane protein hydrophobicity causes a distortion in the distribution of bilayer phospholipids due to their tendency to match the hydrophobic surface of the protein (*Figure 1A*). This not only reduces the membrane thickness near the aqueduct but also produces a bend in the bilayer centered on the aqueduct so that the membrane is high on the TM4 side and low on the TM6 side of the aqueduct. To quantify the membrane deformation, the number and distribution of phospholipid phosphorus atoms within 10 Å of the protein that spontaneously penetrate the central core of the membrane were measured during the simulation (*Figure 1B,C*). At t = 0 the phosphate groups are concentrated in two main peaks corresponding to the inner and the outer leaflets. As the simulation continues, the leaflets become deformed, as indicated by a broader distribution of phosphorous atoms. Because the rate of diffusion of a molecule across the membrane is inversely related to the distance (Fick's Law), membrane thinning will prime lipid translocation by reducing the energy required to move a lipid across the membrane. Membrane bending may also favor non-bilayer phases that facilitate scrambling.

### The lipid translocation pathway

To gain a detailed description of lipid-protein interactions during scrambling, we characterized the behavior of lipids in the immediate proximity of the aqueduct. Spontaneous penetration of lipid head groups into the membrane interior along the aqueduct was observed, resulting in a continuous file of lipids connecting the outer and inner leaflets (*Figure 2A* middle and right panels; *Figure 2—figure supplement 1*). Each aqueduct contained an average of 3.3 phospholipid phosphates in its central 20 Å-thick core and accommodated up to six phosphates (*Figure 2*). During the last 500 ns of the simulation, the aqueduct was occupied by >4 phosphates ~40% of the time (*Figure 2C*). The structure had converged on a stable conformation after 150 ns of the simulation and exhibited a heavy-atom RMSD of 3.6 ± 0.4 Å for the transmembrane domains compared to the crystal structure.

Key insights into the mechanisms of phospholipid scrambling were obtained by measuring the dynamics of lipid translocation along the aqueduct (*Figure 3*). At t = 0, lipid head groups in the bilayer clustered around z ~ ±18 Å (*Figure 1C*). During the simulation, lipid head groups from the inner and outer leaflets were quickly funneled towards the aqueduct entrances through wide vestibules to become concentrated near the aqueduct entrances (z ~ ±12 Å) (*Figure 3A*). As lipids

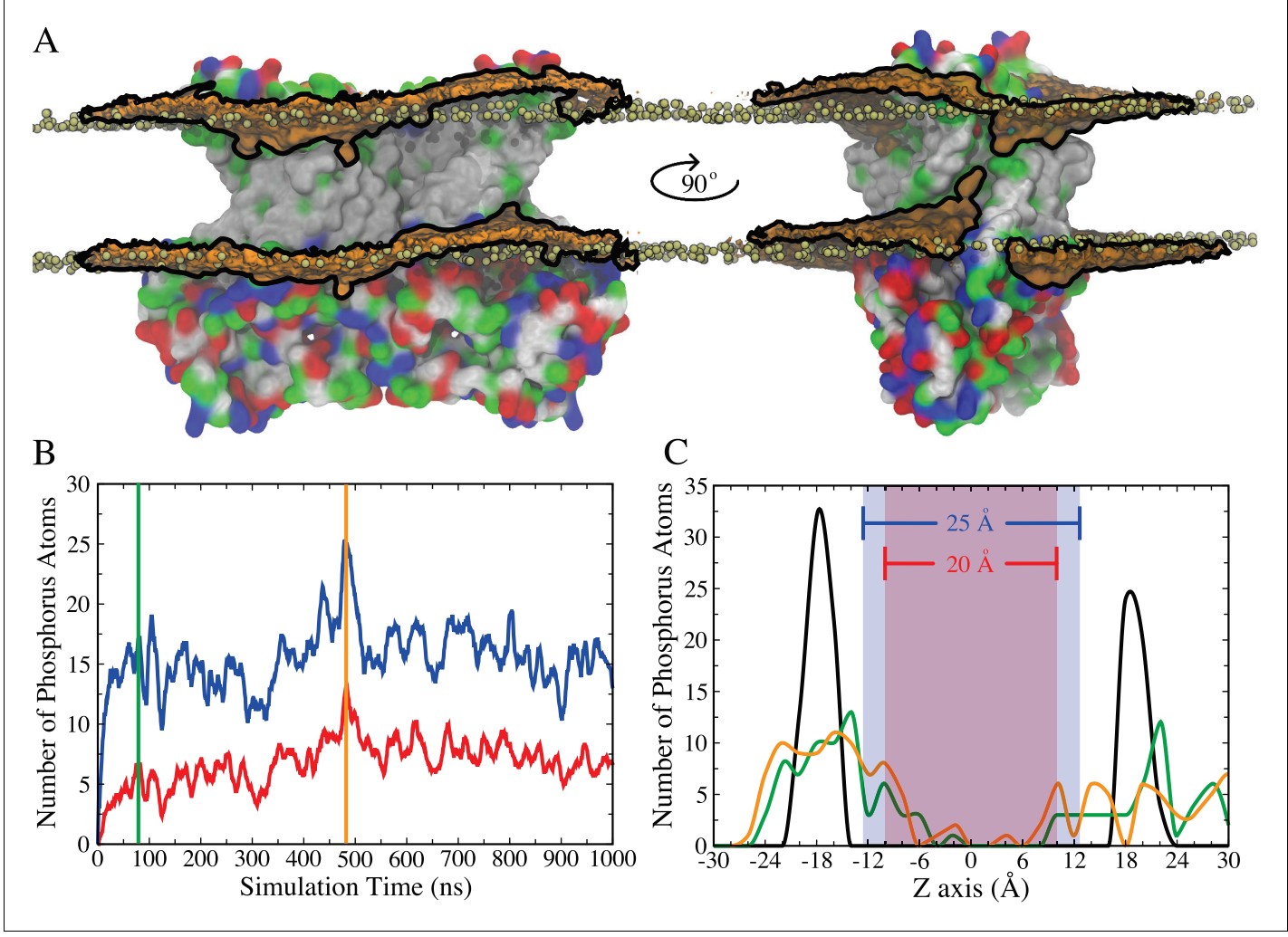

**Figure 1.** Membrane deformation induced by nhTMEM16. (**A**) nhTMEM16 is shown as a molecular surface (extracellular side up) colored by residue type (blue, basic; red, acidic; green, polar; white, nonpolar) in a phospholipid bilayer composed of POPC and POPS. The initial (*t* = 0) distribution of phospholipids is shown by the tan spheres representing lipid phosphorus atoms. At the end of the 1000-ns MD simulation, the average phospholipid phosphate density within 10 Å of the protein during the $Ca^{2+}$-activated simulation is shown as an orange surface outlined by black lines that is contoured at 7.5% of the bulk phosphate density. Two different views (rotated by 90° around the membrane normal) are shown, with one of the aqueducts visible in the right panel. (**B**) Number of phosphorus atoms within 10 Å of the protein that enter the 25 Å- (blue curve) or 20 Å- (red curve) thick core region of the membrane, as defined by the blue and red areas in panel (**C**). (**C**) Distribution of phosphorus atoms along the membrane z axis at *t* = 0 (black curve), *t* = 80 ns (green curve, corresponding to vertical green bar in panel (**B**)), and *t* = 480 ns (orange curve, corresponding to vertical orange bar in panel (**B**)).

DOI: https://doi.org/10.7554/eLife.28671.002

The following figure supplement is available for figure 1:

**Figure supplement 1.** nhTMEM16 surface hydrophobicity.

DOI: https://doi.org/10.7554/eLife.28671.003

migrated along the aqueduct, a third peak was observed near the center of the membrane (z ∼ −3 Å). This site was typically occupied by a single lipid for an extended time (ranging from 350 ns to 900 ns) (*Figure 3A*). On one occasion, transition of a lipid to the extracellular entrance of the aqueduct from this site was observed after it had dwelled at the central site for ∼450 ns (*Figure 3A*, right panel, red trace at 600 ns). Concurrently, the head group at the extracellular entrance was knocked away and quickly merged into the outer leaflet (*Figure 3A* right panel, cyan trace). As this transition was completed, the lipid following in the queue jumped from the intracellular vestibule to the center of membrane (*Figure 3A* right panel, green trace). While this represents a single observation, it

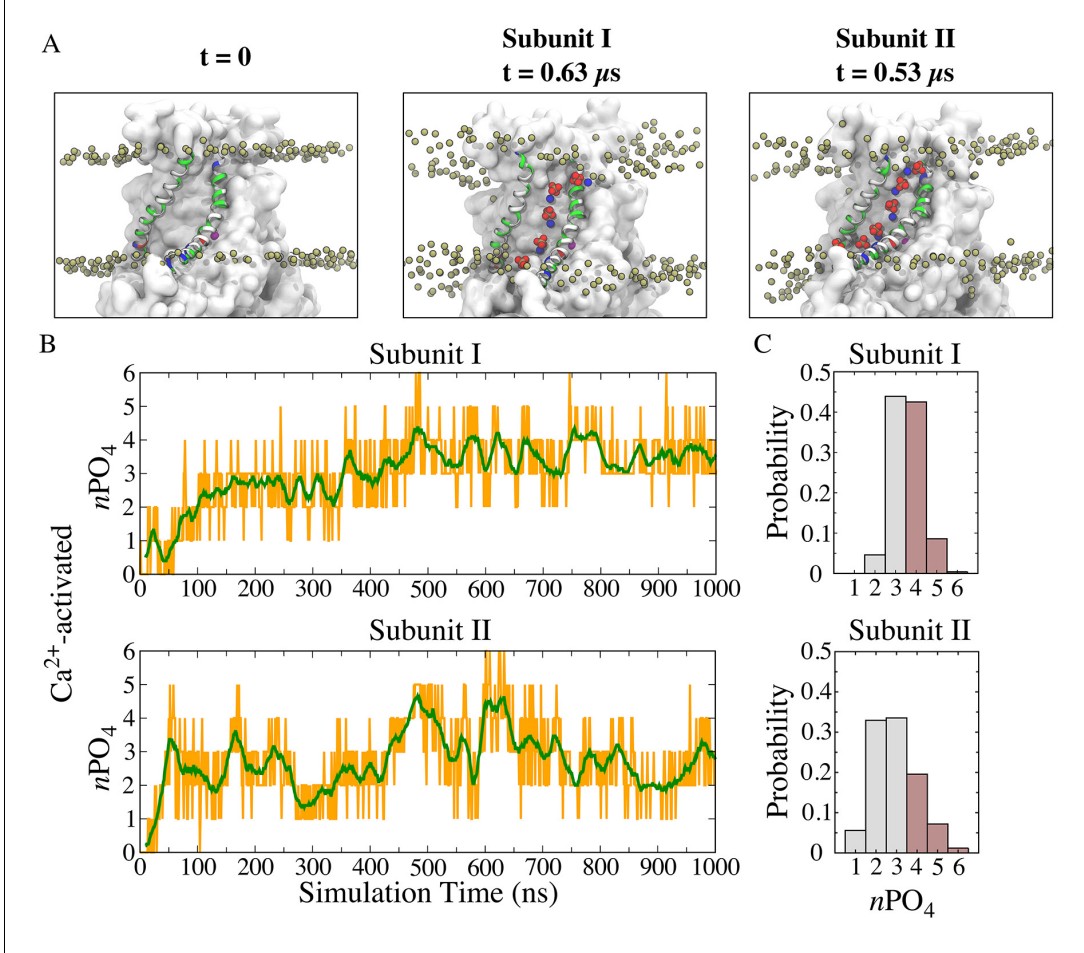

**Figure 2.** Features of the 'aqueduct' on nhTMEM16 surface. (A) Representative snapshots of the aqueduct during an MD simulation with $Ca^{2+}$-activated nhTMEM16. Aqueduct-lining helices TM4 and TM6 are colored by residue type. Phospholipid head groups in the bilayer are shown by tan phosphorus atoms while those inside the aqueduct are shown by phosphorus (tan), oxygen (red) and nitrogen (blue) van der Waals atoms. $Ca^{2+}$ ions are purple spheres. The aqueduct is devoid of phospholipid head groups at $t = 0$ (left panel), but becomes occupied by lipids in both subunits during the simulation. (B) Number of phosphate groups (orange trace) and the moving average (green, bin = 20) within the 20 Å-thick core region. (C) Normalized probability histograms of the phosphate count in the 20 Å-thick core region of each aqueduct during the last 500 ns. Numbers larger than three are highlighted in brown.

DOI: https://doi.org/10.7554/eLife.28671.004

The following figure supplement is available for figure 2:

**Figure supplement 1.** Penetration of lipids into the aqueduct in the $Ca^{2+}$-activated simulation.

DOI: https://doi.org/10.7554/eLife.28671.005

suggests that lipids move along the aqueduct in a highly concerted fashion. Notably, the lipid that translocated from the inner leaflet remained near the extracellular entrance for a few hundred nanoseconds (650 to 945 ns) before eventually diffusing away from the aqueduct and merging into the outer leaflet (not shown) (*Figure 3—video 2*, final 2 s). This event represented a full permeation of the lipid from the inner leaflet to the outer leaflet (*Figure 3—videos 1* and *2*). While we observed only one full translocation under conditions of zero voltage across the bilayer, we have observed four more full translocations when voltage was applied as described later.

While phospholipid head groups were wedged in the aqueduct, their flexible hydrocarbon tails remained in the hydrophobic phase of the membrane (*Figure 3B*). Hydrophobic interactions between the lipid tails and the aqueduct-lining helices (TM4 and TM6) were also frequently observed. TM4 and TM6 are amphipathic with hydrophilic residues facing mainly toward the inside of the aqueduct and hydrophobic residues exposed to the bilayer interior. This structural

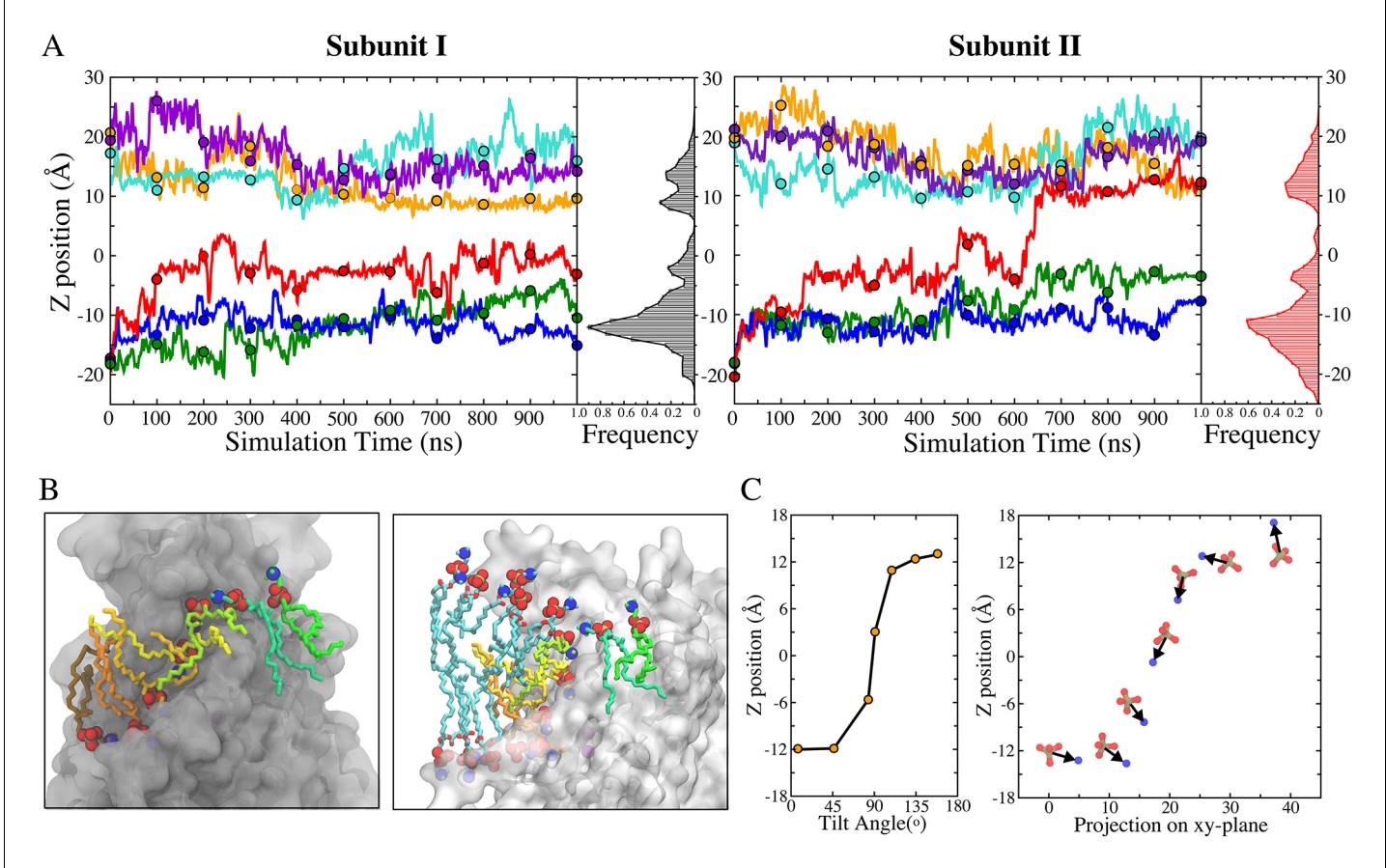

**Figure 3.** Kinetics of lipid permeation. (**A**) Translocation of lipids along the aqueduct measured as the z-positions of phosphorus atoms versus time. Each 100-ns time point is highlighted as a dot. Complete flipping of a lipid from the inner leaflet to the outer leaflet is captured in subunit II (red trace). Histograms plot the frequency of phosphorus atoms in 1-Å bins along the aqueduct. (**B**) Views showing representative lipids in the aqueduct. Left panel: Front view with hydrocarbon chains of each phospholipid inside the aqueduct colored differently. Right panel: same image rotated 90° with added phospholipids in proximity to the aqueduct colored with cyan tails. The head groups of these lipids are organized in a curve with the lipids closest to the aqueduct being located closer to z = 0. (**C**) Left: representative orientations of lipid tails along the aqueduct. The tilt angle of the lipid tail is defined by the angle bisector vector of the two acyl tails and z axis. Right: corresponding P-N dipole orientations of the head groups along the aqueduct.

DOI: https://doi.org/10.7554/eLife.28671.006

The following video and figure supplement are available for figure 3:

**Figure supplement 1.** Fraction of time each amino acid forms hydrophobic contact (<4 Å) with a fatty acid acyl chain whose head group is inside the aqueduct.

DOI: https://doi.org/10.7554/eLife.28671.007

**Figure 3—Video 1.** Full scrambling of a POPC lipid from inner leaflet to outer leaflet of the membrane through the aqueduct under equilibrium condition.

DOI: https://doi.org/10.7554/eLife.28671.008

**Figure 3—Video 2.** Full scrambling of a POPC lipid from inner leaflet to outer leaflet of the membrane through the aqueduct under equilibrium condition.

DOI: https://doi.org/10.7554/eLife.28671.009

arrangement allows lipid tails to splay out and make contacts with the lining helices, while the head group is wedged in the aqueduct (*Figure 3B* left panel, *Figure 3—figure supplement 1*). Lipids were also commonly observed to have one alkyl tail attached to the hydrophobic surface of the lining helices and the other tail extending into the hydrophobic core of the membrane as it moved along the aqueduct (*Figure 3B* left panel). In the one complete translocation that we observed, the lipid tails underwent ~170 degrees of rotation as the head group shifted from one leaflet to the other along the aqueduct (*Figure 3C*).

## Pinpointing residues controlling scrambling

Although locations of the single file lipid phosphate groups within the aqueduct fluctuate, the most visited sites were identified by calculating the density of the phosphate groups over the trajectory as well as determining the electrostatic interactions between the head groups and protein atoms (*Figure 4A* left panel). The fraction of simulation time that each residue forms electrostatic interactions with a lipid head group inside the aqueduct is shown in *Figure 4—figure supplement 1*. The three most visited sites are named 'internal site ($S_{int}$)', 'central site ($S_{cen}$)', and 'external site ($S_{ext}$)'. $S_{int}$ is located at a z-level similar to the lower bound $Ca^{2+}$ ion. The phosphate group at $S_{int}$ is in direct contact with Q374 and N378 (TM5), and R505 (TM7). Notably, R505 sits between two negatively charged residues, E506 and D503, which are among the most highly conserved residues in the TMEM16 family (ranking 9/9 by a multiple sequence alignment of 392 sequences from 38 phylogenetic classes by CONSURF (*Ashkenazy et al., 2010*)) and contribute to coordinating the bound $Ca^{2+}$ ions. $S_{cen}$, located about one third into the membrane thickness from the intracellular side, contributes coordinating residues N378, T381, S382 (TM5), and T340 (TM4). $S_{ext}$ is located near the extracellular entrance of the aqueduct and is comprised of several hydrophilic residues including E313 and N317 (TM3), K325, Q326, T333 (TM4), and R432, N435, and Y439 (TM6).

To test whether amino acids that interact with lipid head groups in the simulation are crucial in phospholipid scrambling, mutations were introduced into selected lipid-coordinating residues whose physicochemical properties were conserved between the scramblases nhTMEM16 and TMEM16F but were different in the Cl⁻ channel TMEM16A. We focused on TM4 and TM5 because we had previously shown that amino acids in this region in TMEM16F are critical for phospholipid scrambling (*Yu et al., 2015*). Of the 14 amino acids in TM4 –TM5 that are involved in direct interaction with lipid head groups (filled circles under the alignment in *Figure 4B*), two amino acids (consensus positions 333 and 378, highlighted in red) were targeted for mutagenesis, since they are similar in nhTMEM16 and TMEM16F but distinctly different in TMEM16A. To improve nhTMEM16 expression, a codon-optimized nhTMEM16 cDNA was synthesized. Although nhTMEM16 tagged on the C-terminus with EGFP did not traffic to the plasma membrane (*Brunner et al., 2014*), nhTMEM16 tagged on the N-terminus with 3X-FLAG or untagged nhTMEM16 did traffic to the plasma membrane (*Figure 4C*). Two nhTMEM16 mutants were generated by mutating T333 and N378 to the amino acids corresponding to mouse TMEM16A (T333V and N378K). Intracellular $Ca^{2+}$ was elevated by whole-cell patch clamping with an intracellular solution containing 200 μM free $[Ca^{2+}]$. Phospholipid scrambling was assessed by imaging the accumulation of the phosphatidylserine probe Annexin-V-AlexaFluor-568 on the cell surface. Most cells expressing WT nhTMEM16 had scrambled by 15 min after elevating cytosolic $Ca^{2+}$ by establishing whole cell recording. This slow time course of scrambling is similar to that previously described for TMEM16F (*Yu et al., 2015*). It is explained partly by the sensitivity of the Annexin-V binding assay, but apparently also involves a regulatory step we do not understand. No scrambling was observed with WT nhTMEM16 expressing cells patch clamped with zero $Ca^{2+}$ during the same time period. This result differs from that reported for purified fungal TMEM16 proteins reconstituted in liposomes that show substantial scramblase activity in the absence of $Ca^{2+}$ (*Malvezzi et al., 2013*; *Brunner et al., 2016*). Less than 20% of untransfected cells scramble under similar conditions, as we reported previously (*Yu et al., 2015*). While this has not been explored systematically, untransfected HEK cells express low levels of TMEM16F that may be responsible for scrambling in a small fraction of cells.

In contrast to the robust scrambling observed with WT nhTMEM16 in the presence of $Ca^{2+}$, very little scrambling was observed in cells expressing T333V or N378K mutants of nhTMEM16 (*Figure 4D–F*). This supports the conclusion that these residues are important in the scrambling mechanism. In addition to these residues in TM4 and TM5, we also mutated R505 in TM7 because its charge could potentially interact differently with charged and uncharged phospholipids. The R505Q mutation greatly reduced phospholipid scrambling (*Figure 4D–F*).

## Ionic currents associated with lipid scrambling

It has been reported that the fungal TMEM16 scramblases nhTMEM16 (*Brunner et al., 2014*; *Lee et al., 2016*) and afTMEM16 (*Malvezzi et al., 2013*) as well as the mammalian scramblase TMEM16F (*Yu et al., 2015*) (and references therein) conduct ions as well as scramble lipids (*Whitlock and Hartzell, 2016*). To investigate the relationship between ion conduction and lipid

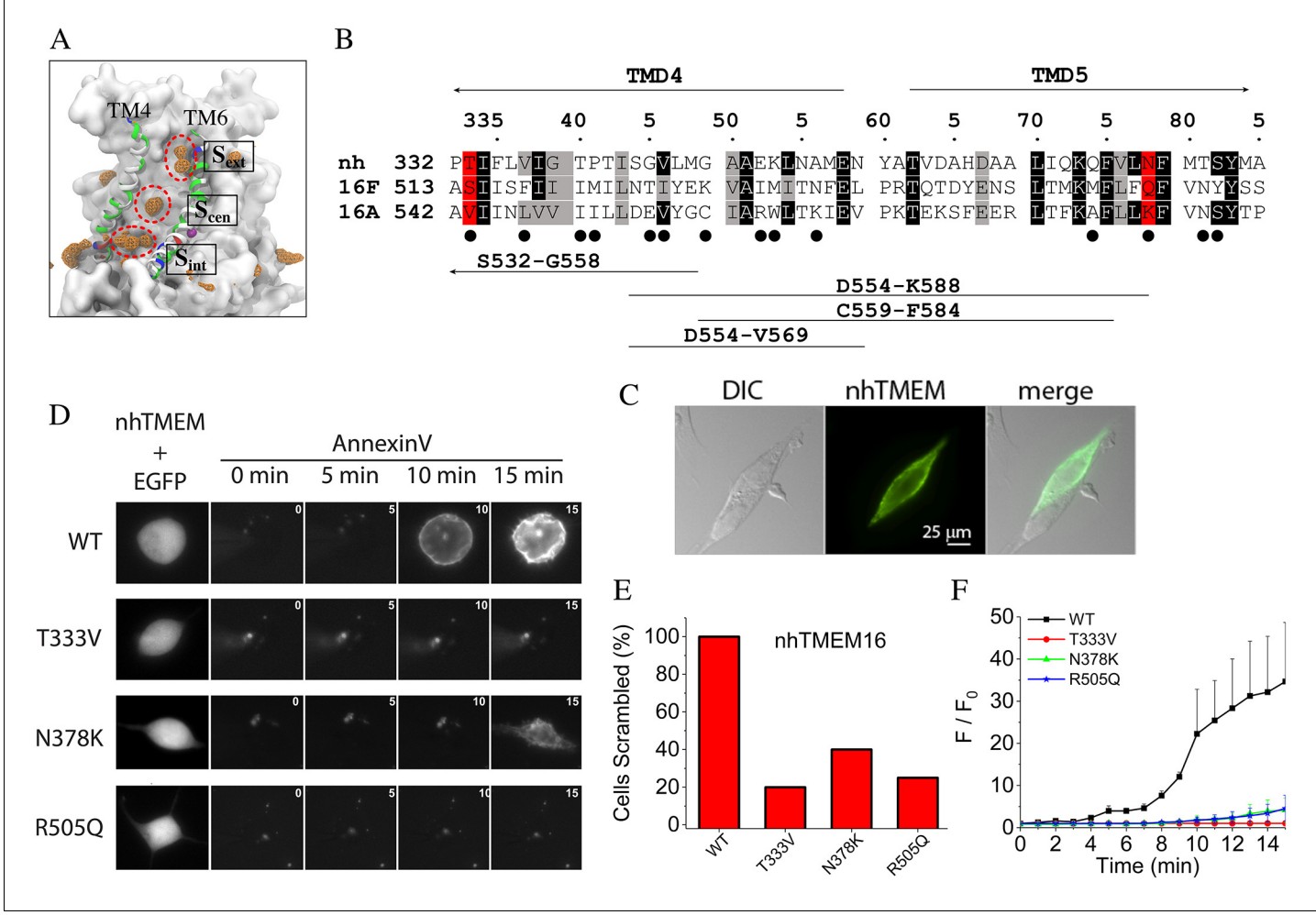

**Figure 4.** Lipid binding sites and effects of mutations in nhTMEM16. (A) Volumetric map of phosphate occupancy extracted from the Ca²⁺-activated simulation is shown as orange wireframe contoured at isovalue 0.15 overlaid on the protein structure. The three phosphate binding sites (circled and labeled) are characterized in detail in *Figure 4—Figure supplement 1*. (B) Alignments of TM4 and TM5 of nhTMEM16 (nh), mouse TMEM16F (16F), and mouse TMEM16A (16A). Structure-based alignments were performed using PROMALS3D. The sequences of nhTMEM16 and the mammalian proteins are 31% similar over the aligned region. Backgrounds are colored to indicate identity (black) or similarity (grey). nhTMEM16 amino acids that directly interact with lipid head groups for >10% of the time are indicated with a filled circle (•). Amino acids that interact with lipid head groups and are conserved between nhTMEM16 and TMEM16F but different in TMEM16A are highlighted in red. Consensus amino acid numbering in the top line refers to nhTMEM with the first residue numbers preceding the sequence. At the bottom, the four horizontal lines indicate TMEM16A sequences that were previously replaced with TMEM16F sequences to convert TMEM16A into a phospholipid scramblase (*Yu et al., 2015*). (C) 3X-FLAG-nhTMEM16 is trafficked to the plasma membrane of HEK-293 cells. Cells were transiently transfected with a synthetic codon-optimized nhTMEM16 tagged on the N-terminus with 3X-FLAG. Cells were fixed and stained with anti-FLAG antibody followed by a fluorescent secondary antibody and imaged with a wide-field microscope. *Left:* differential interference contrast image. *Middle:* immunofluorescence. *Right:* merged image. Two attached cells are shown. One cell was transfected with nhTMEM16 and one was not transfected. (D) *Left:* effects of mutations in nhTMEM16 on phospholipid scrambling. Cells were co-transfected with plasmids encoding nhTMEM16 (WT or mutant) and EGFP. Transfected cells were identified by green fluorescence (left panel). Phospholipid scrambling was measured by binding of Annexin-V-AlexaFluor-568 at 0, 5, 10, and 15 min after establishing whole-cell recording with an internal solution containing 200 µM free Ca²⁺ (right panels). (E) Percentage of cells binding Annexin-V above background within 15 min after establishing whole cell recording with a high Ca²⁺ internal solution. Each bar represents 4 to 5 cells. (F) Time course of phospholipid scrambling by WT and mutant nhTMEM16.

DOI: https://doi.org/10.7554/eLife.28671.010

The following figure supplement is available for figure 4:

**Figure supplement 1.** Amino acid residue interaction with lipid head groups in the aqueduct.

DOI: https://doi.org/10.7554/eLife.28671.011

scrambling by nhTMEM16, ionic currents and lipid scrambling were measured simultaneously in HEK cells transiently transfected with WT or mutant nhTMEM16 in response to voltage steps. nhTMEM16 elicited phospholipid scrambling and ionic currents when intracellular $Ca^{2+}$ was elevated (*Figure 5A*). Elevation of cytosolic $Ca^{2+}$ by establishing whole-cell recording with high $Ca^{2+}$ in the patch pipet resulted in exposure of phosphatidylserine over 5–10 min that was accompanied by a large ionic current.

WT nhTMEM16 ionic current activated by 200 µM free $Ca^{2+}$ exhibited a linear current-voltage relationship that reversed at ~0 mV under symmetrical ionic conditions. To determine ionic selectivity, we measured the change in reversal potential ($E_{rev}$) in response to reducing extracellular NaCl (or CsCl) from 150 mM to 30 mM. Upon 10-fold reduction in NaCl, $E_{rev}$ shifted −7 mV (*Figure 5B*), which corresponds to $P_{Na}:P_{Cl} = 2.3 \pm 0.2$ as calculated by the Goldman, Hodgkin, Katz equation (*Figure 5 C*). $P_{Cs}:P_{Cl}$ was $1.5 \pm 0.1$. Therefore, the currents associated with nhTMEM16 scrambling are slightly cation-selective. Our suggestion that ionic currents are linked to phospholipid scrambling is supported by the observation that ionic currents were greatly reduced in cells expressing nhTMEM16 mutants T333V, N378K, and R505Q that exhibit reduced scrambling (*Figure 5 D*).

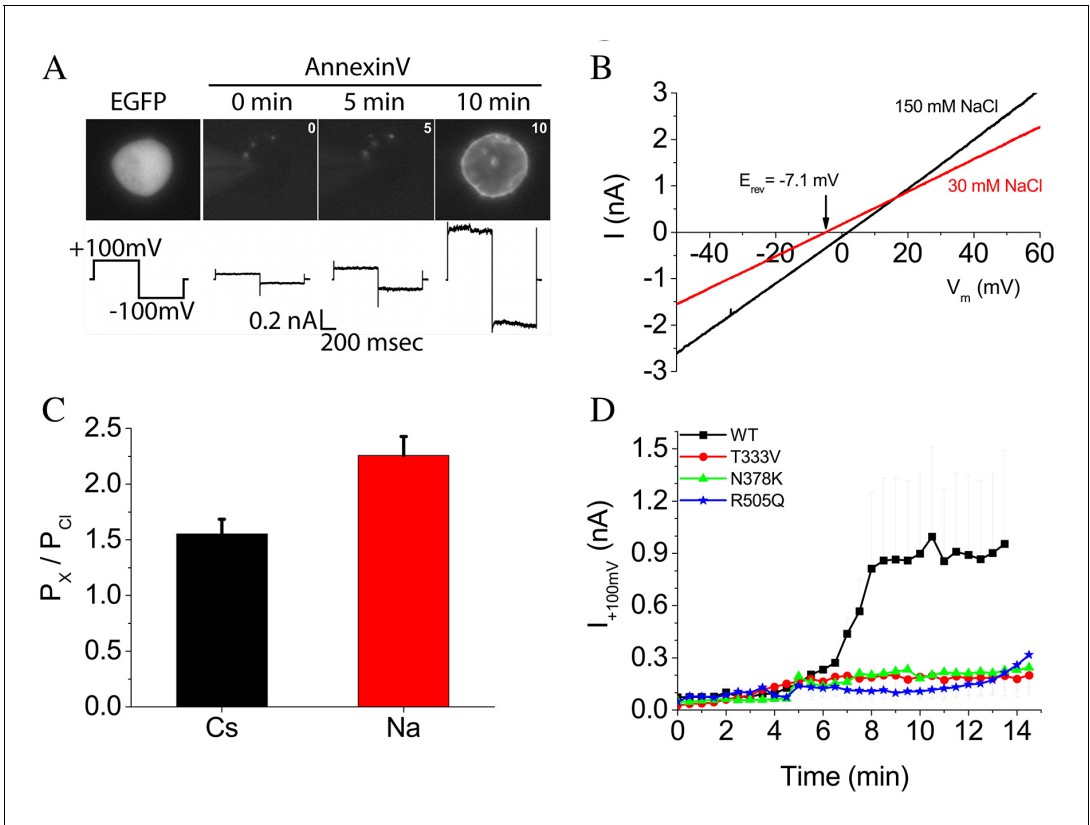

**Figure 5.** Phospholipid scrambling and ionic currents in nhTMEM16. (**A**) Phospholipid scrambling and ionic currents are stimulated in HEK cells transfected with nhTMEM16 and co-transfected with a plasmid encoding EGFP. Transfected cells were identified by green fluorescence. Phospholipid scrambling was measured by binding of Annexin-V-AlexaFluor-568 at 0, 5, and 10 min after establishing whole-cell recording with an internal solution containing 200 µM free $Ca^{2+}$. Ionic currents recorded by a voltage step from a holding potential of 0 mV to +100 mV followed by a step to −100 mV are shown below each Annexin-V fluorescent image. Untransfected HEK cells do not scramble in the 15 min time frame as previously shown (*Yu et al., 2015*). (**B**) Ionic selectivity of WT nhTMEM16 currents. Current-voltage relationships were measured using voltage ramps 1 s duration from −100 mV to +100 mV. Ionic selectivity was determined by the change in reversal potential measured in response to switching between extracellular solutions containing 150 mM NaCl (black curves) and 30 mM NaCl (red curves). (**C**) Relative permeability of $Na^+$ and $Cs^+$ to $Cl^-$ ($P_{Na}:P_{Cl}$ and $P_{Cs}:P_{Cl}$) calculated from the Goldman-Hodgkin-Katz equation. (**D**) Time course of development of ionic currents at +100 mV by WT and mutant TMEM16.
DOI: https://doi.org/10.7554/eLife.28671.012

## Testing the model on mammalian TMEM16F

To test whether these residues are also important in phospholipid scrambling mediated by the mammalian scramblase TMEM16F, we mutated the amino acids in TMEM16F homologous to T333 and

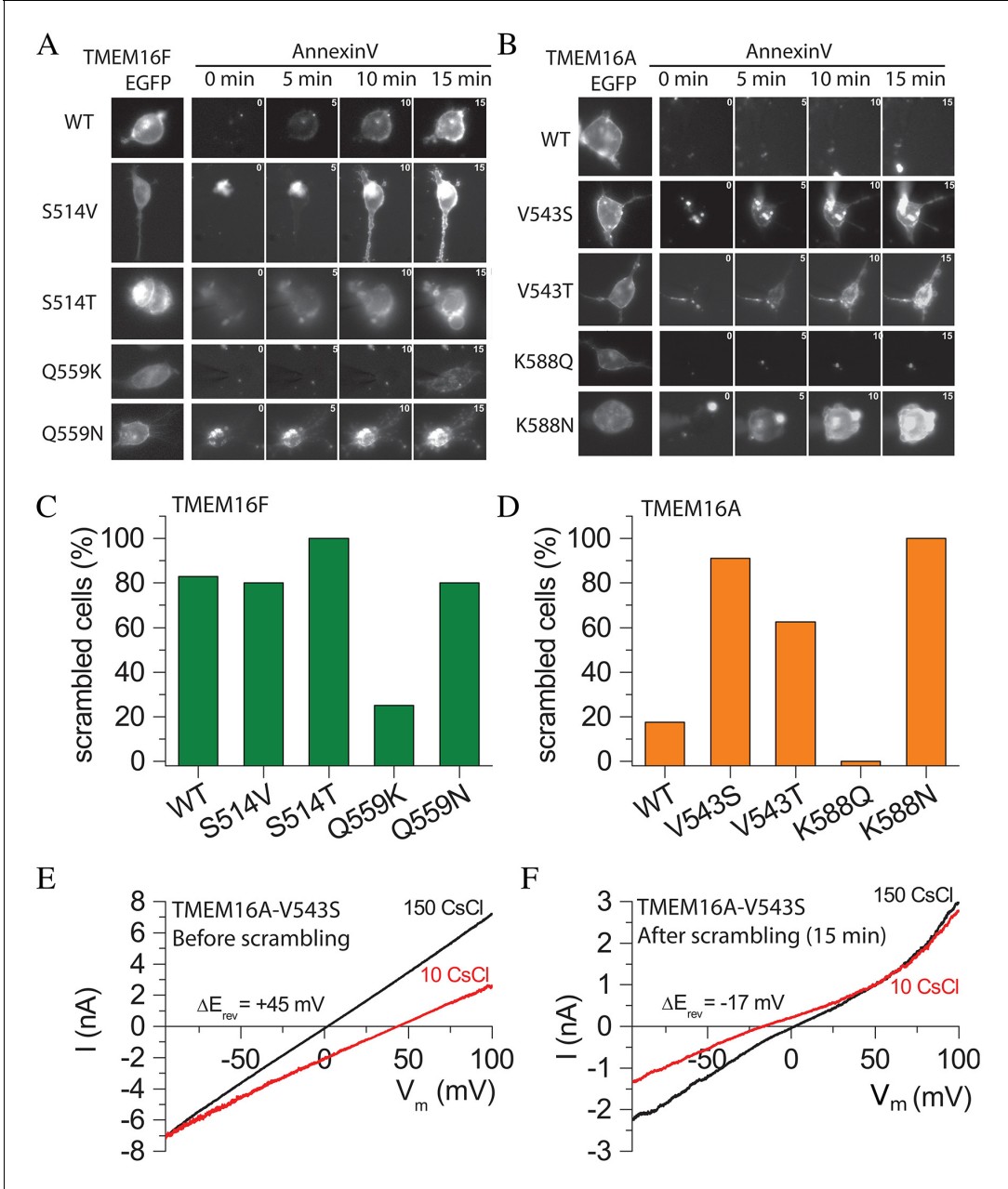

**Figure 6.** Effects of amino acid mutations on TMEM16F (**A,C**) and TMEM16A (**B,D**). (**A and B**) Images of cells during phospholipid scrambling. Single cells were patch clamped and intracellular $Ca^{2+}$ controlled by 200 μM $Ca^{2+}$ in the pipet. The first image in each row is EGFP fluorescence of the tagged TMEM16 protein. The next images are Annexin-V fluorescence taken 0, 5, 10, and 15 min after raising $Ca^{2+}$ by initiating whole cell recording. (**C and D**) Percentage of cells binding Annexin-V within 15 min after establishing whole cell recording with a high $Ca^{2+}$ internal solution. Each bar represents between 5 and 29 cells. (**E and F**) The V543S mutation in TMEM16A alters ionic selectivity. Current-voltage relationships were measured using voltage ramps of 1 s duration from −100 mV to +100 mV. Ionic selectivity was determined by the difference in reversal potential between extracellular 150 mM CsCl (black curves) and 10 mM CsCl (red curves). (**E**) Switching between 150 mM and 10 mM CsCl immediately after establishing whole-cell recording shifted $E_{rev}$ +45 mV, corresponding to a $P_{Cl}:P_{Cs}$ = 9.7, whereas (**F**) switching after Annexin-V binding had reached a plateau, $E_{rev}$ shifted −17 mV in the opposite direction, corresponding to $P_{Cl}:P_{Cs}$ = 0.45.

DOI: https://doi.org/10.7554/eLife.28671.013

N378 in nhTMEM16 (*Figure 6A and C*). As previously reported (*Yu et al., 2015*), >80% of HEK cells expressing WT TMEM16F exhibited Annexin-V binding within 15 min of establishing whole-cell recording with a high $Ca^{2+}$ intracellular solution (*Figure 6C*) and also exhibited robust non-selective currents that developed roughly in parallel with Annexin-V binding. When TMEM16F Q559 (homologous to nhTMEM16 N378) was mutated to the corresponding TMEM16A residue (Q559K), scrambling and ionic currents were greatly reduced (*Figure 6C*). As expected, replacing TMEM16F Q559 with the corresponding amino acid from nhTMEM16 (Q559N) did not abrogate scrambling by TMEM16F (*Figure 6C*). These data confirm that the TMEM16F residue homologous to nhTMEM16 N378 is important in phospholipid scrambling. Mutating S514 in TMEM16F to the corresponding nhTMEM16 amino acid (S514T) had no effect on scrambling, as expected. We initially expected that replacing TMEM16F S514 with V from TMEM16A would abrogate scrambling, but the S514V mutation had little effect on scrambling. Recently a low-resolution structure of TMEM16A was published (*Paulino et al., 2017*) that provides additional insight into these results (see Discussion).

## Point mutations confer scramblase activity on TMEM16A

Previously we reported that replacing short TMEM16A sequences in TM4–TM5 with corresponding TMEM16F sequences caused TMEM16A to scramble phospholipids robustly, while WT TMEM16A exhibited no scramblase activity (*Yu et al., 2015*). We therefore asked whether TMEM16A could be converted to a scramblase by mutating V543 to the homologous scramblase amino acids (T in nhTMEM16 or S in TMEM16F). Fewer than 20% of the WT TMEM16A-expressing cells exhibited scramblase activity within 15 min of cytosolic $Ca^{2+}$ elevation (*Figure 6B and D*), but cells expressing TMEM16A mutants V543T or V543S exhibited robust phospholipid scrambling in the same time period (*Figure 6D*), consistent with the hypothesis that this amino acid plays a key role in conducting lipids across the bilayer. Replacing TMEM16A K588 with the corresponding nhTMEM16 amino acid N (K588N) also induced robust phospholipid scrambling (*Figure 6D*). However, replacing K588 with the corresponding TMEM16F amino acid Q did not induce scrambling. One possible explanation is that the side chain of N is considerably shorter (3.7 Å) than Q (4.9 Å) or K (6.2 Å). Q and K side chains may be long enough to interact with nearby amino acids to stabilize the $Cl^-$ channel conformation of TMEM16A, while N may favor a scramblase conformation because it cannot make these interactions. The ability to convert TMEM16A into a phospholipid scramblase by point mutations in each of two major amino acids identified as interacting with translocating lipids in the MD simulation supports the validity of the MD model. Additionally, this result provides support for the idea that the TMEM16 $Cl^-$ channels and scramblases are structurally and functionally very similar.

In addition to conferring scramblase activity on TMEM16A, the V543S mutation also changed the ionic selectivity of TMEM16A (*Figure 6E and F*). Immediately after initiating whole-cell recording, TMEM16A V543S currents were strongly $Cl^-$ selective, because switching from extracellular 150 mM CsCl to 10 mM CsCl caused a +45 mV shift of $E_{rev}$ corresponding to $P_{Cl}/P_{Cs}$ = 9.7 (*Figure 6E*). With time, the reversal potential changed as phospholipid scrambling developed. The ionic current became weakly cation-selective as evidenced by a −17 mV negative shift in $E_{rev}$ upon switching from 150 mM CsCl to 10 mM CsCl. This corresponds to a 21-fold decrease in anion selectivity ($P_{Cl}/P_{Cs}$ = 0.45) (*Figure 6F*). Thus, the V543S mutant of TMEM16A appears to function initially as a $Cl^-$ channel and then becomes less selective as scrambling is activated.

## The aqueduct also conducts ions in simulations

To explore the relationship between ionic currents and phospholipid scrambling, we performed additional MD simulations under conditions where various voltages were applied across the membrane (−150 mV, −250 mV, and −500 mV). With the imposition of transmembrane potentials, the number of lipid head groups in the central 20 Å-thick core of membrane increased from 3.3 at 0 mV to 3.4 at −150/−250 mV, and 4.0 at −500 mV. Further, $Na^+$ permeation from the extracellular side to the intracellular side of the membrane was observed to occur through the aqueduct. At −150 mV and −250 mV, $Na^+$ permeation events occurred in both subunits during the 700 ns simulations. The average coordination number of permeant $Na^+$ ions during the course of ion translocation was 5.70 ± 0.51 (*Figure 7A*). Coordination is supplied mainly by oxygen atoms from water molecules (83.4%). The remainder was split between protein (62.5%) and lipid (37.5%) (*Figure 7A*). $Na^+$ ions dwell for an extended time at two locations (*Figure 7A* black curves, *Figure 7—figure supplement*

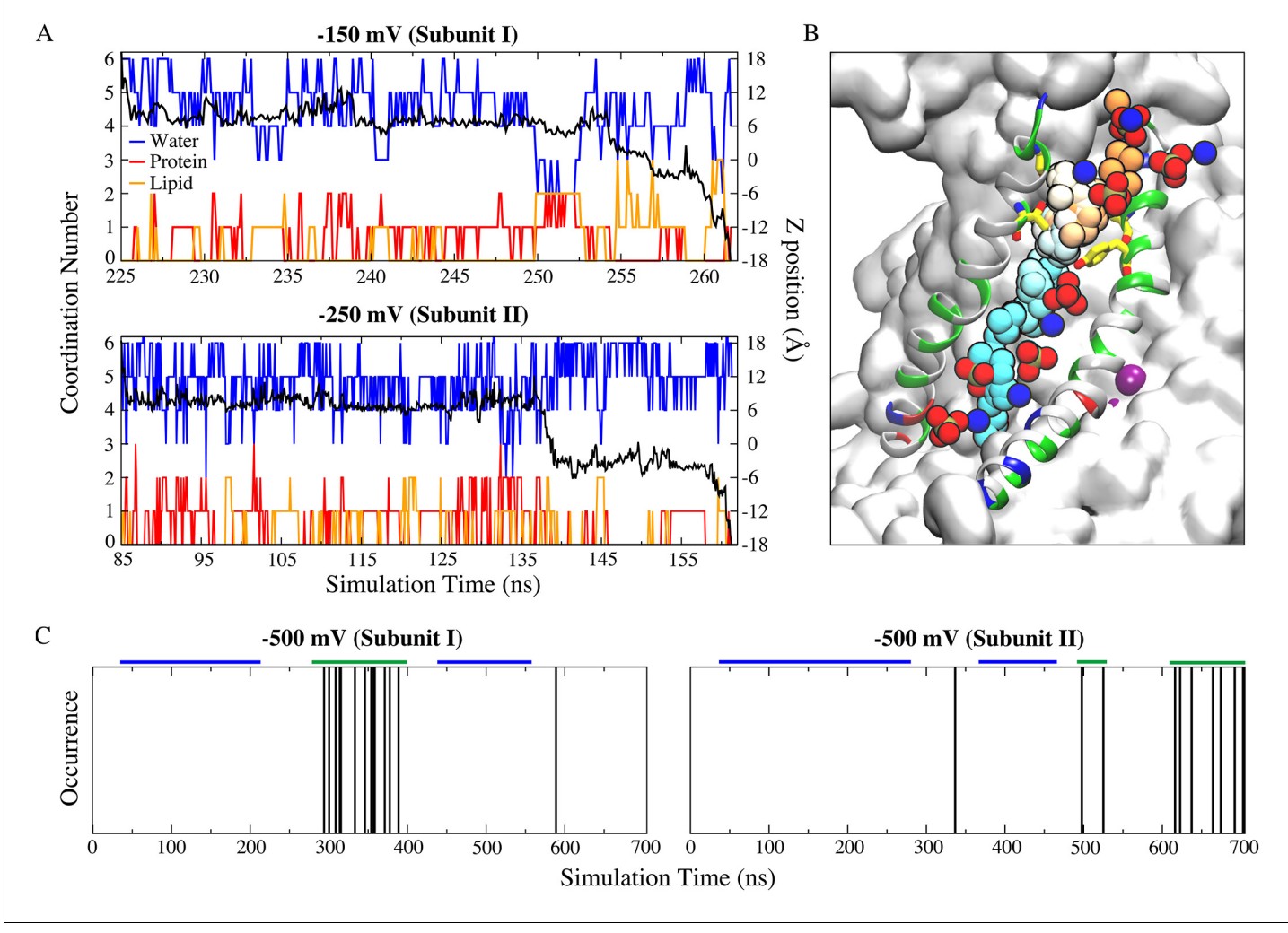

**Figure 7.** Ion permeation occurs through the same structural pathway of lipid scrambling. (A) Full permeation of one $Na^+$ ion from the extracellular side to the intracellular side of the membrane at −150 mV (top panel) and −250 mV (bottom panel). Coordination number (left y-axis) and z-position (right y-axis) of the permeant $Na^+$ ions as a function of time. Oxygen atoms from protein (red) and lipid head groups (orange) both contribute to the ion coordination throughout their permeation (−150 mV: ~36.7 ns duration starting at ~225 ns; −250 mV: ~76.4 ns duration starting at ~85 ns). The coordination number for water oxygen atoms is blue. The black curve is the corresponding z-position of the permeant ion. (B) Time series snapshots representing translocation of the $Na^+$ ion with time represented in pseudo-color from orange (start) to cyan (end). Y439 and T333 are shown in stick representations. The lipid head groups lining the ion permeation pathway are shown as phosphate groups and nitrogen atoms. (C) Multiple ion permeation events occur in both subunits at −500 mV during the 700 ns simulation. Each ion permeation event is indicated by a vertical line at the time point when the permeation is complete. The segments marked by the blue and green bars were used for measuring the center of mass distances between the aqueduct lining helices TM4 and TM6 as shown in *Figure 7—figure supplement 4*.

DOI: https://doi.org/10.7554/eLife.28671.014

The following video and figure supplements are available for figure 7:

**Figure supplement 1.** Translocation of the permeant $Na^+$.

DOI: https://doi.org/10.7554/eLife.28671.015

**Figure supplement 2.** $Na^+$ permeation at −500 mV.

DOI: https://doi.org/10.7554/eLife.28671.016

**Figure supplement 3.** At higher voltages (−1 V), both $Na^+$ influx (38 occurrences) and $Cl^-$ efflux (eight occurrences) were observed during a 90 ns simulation.

DOI: https://doi.org/10.7554/eLife.28671.017

**Figure supplement 4.** Dilation of the aqueduct.

DOI: https://doi.org/10.7554/eLife.28671.018

**Figure supplement 5.** Comparison of the aqueduct conformation with different transmembrane potentials.

DOI: https://doi.org/10.7554/eLife.28671.019

*Figure 7 continued on next page*

*Figure 7 continued*
**Figure 7—Video 1.** Movie shows part of the simulation trajectory (~200 to 400 ns) at −500 mV, during which multiple ion permeations were captured through the aqueduct in subunit I.
DOI: https://doi.org/10.7554/eLife.28671.020

1A and B, left panels). One $Na^+$ interaction site is located near the $S_{ext}$ site (*Figure 4A*) where the ion is coordinated by oxygen atoms from N310, E313, G329, T333, Q436, and Y439 (*Figure 7—figure supplement 1*). The other site is near the $S_{cen}$ site (*Figure 4A*) involving residues T381, S382, N448, and Y513 (*Figure 7—figure supplement 1*). Q374 from $S_{int}$ and E496 and E497 near the intracellular entrance to the aqueduct also interact transiently with $Na^+$ ions as they leave the aqueduct. Intriguingly, T333, which is a key gating residue for lipid scrambling, is the residue that interacts most extensively (with a time percentage of 36.8%) with permeant $Na^+$ ions. The finding that $Na^+$ interacts with both protein and lipid during its translocation (*Figure 7A and B*) supports the idea that ion transport in scramblases occurs via a 'proteolipidic' pore along the hydrophilic aqueduct during phospholipid scrambling. In this scheme, lipids play a structural role by lining the hydrophilic ion conduction pathway with their head groups. Involvement of lipids in ion translocation pathways has also recently been suggested for the P2X$_3$ receptor (*Mansoor et al., 2016*).

The probability of $Na^+$ permeation increased with voltage. At −500 mV there were 24 $Na^+$ influx events over 700 ns (*Figure 7C*, *Figure 7—figure supplement 2*, *Figure 7—video 1*) and at −1000 mV there were 38 $Na^+$ influx and 8 $Cl^-$ efflux events over 90 ns of simulation (*Figure 7—figure supplement 3*). Importantly, the aqueduct as measured by the center-of-mass (COM) distance between TM4 and TM6 dilates during the frequent ion permeation phase in the simulation (*Figure 7C*, *Figure 7—figure supplement 4*). These large non-physiological voltages were used to increase the frequency of permeation events during the 700 ns simulations by increasing the driving force for ion flux. To evaluate whether the large voltages distorted the protein conformation directly, the TM4-TM6 COM distance under transmembrane potentials were compared to that of the crystal structure and the simulation at 0 mV (*Figure 7—figure supplement 5*). At −150 mV and −250 mV, the TM4-TM6 COM distances are very similar to those in the crystal structure and simulation at 0 mV (*Figure 7—figure supplement 5*); the average COM distances in simulation at −500 mV is slightly larger, due to the transient dilation phase; the 'normal state', which takes the major length of the simulation, is very similar to that in the crystal structure (*Figure 7—figure supplement 4A*).

## POPS externalization and lipid selectivity

To analyze lipid scrambling in the presence of a transmembrane voltage, a combined simulation trajectory with a total aggregate time of 1700 ns (1000 ns at 0 mV, followed by 700 ns at −500 mV) was used. Five full lipid permeation events (2 POPS and 3 POPC) from the inner leaflet to the outer leaflet were observed (*Figure 8A and B*, *Figure 8—figure supplement 1A*). In addition, 3 half-flopping (2 POPC and 1 POPS, from inner leaflet to the membrane mid plane) and 1 half-flipping (1 POPC, from outer leaflet to the membrane mid plane) were observed (*Figure 8—figure supplement 1B*). For the full POPS permeation events, the head groups were well coordinated by protein oxygen and nitrogen atoms (*Figure 8A*). The POPS head group interacts most closely with R505 (59–80% of the scrambling time). E496 and N378 also contribute significantly (>10% of the scrambling time). In contrast, POPC was less coordinated by protein residues. T381, R432, Y439, K459, and R505 interacted most frequently with POPC, but none of these amino acids interacted with the lipid for more than ~10–15% of the transit time across the membrane. A major difference between the scrambling of POPS and POPC through the aqueduct appears to be the amount of time head groups are coordinated by protein during the scrambling process (*Figure 8A and B* bottom panels, *Figure 8C*). The scrambled POPS head group is coordinated by at least two oxygen/nitrogen atoms from the protein an average of ~49% of the time during its transit (*Figure 8C* top panel, *Figure 8—figure supplement 2*); while for POPC, the average probability is only ~5% (*Figure 8C* bottom panel). Three residues inside the aqueduct (E496, R505 and R548) coordinate (coordination ratio >0.2) POPS more strongly than POPC (*Figure 8D*, *Figure 8—figure supplement 3*).

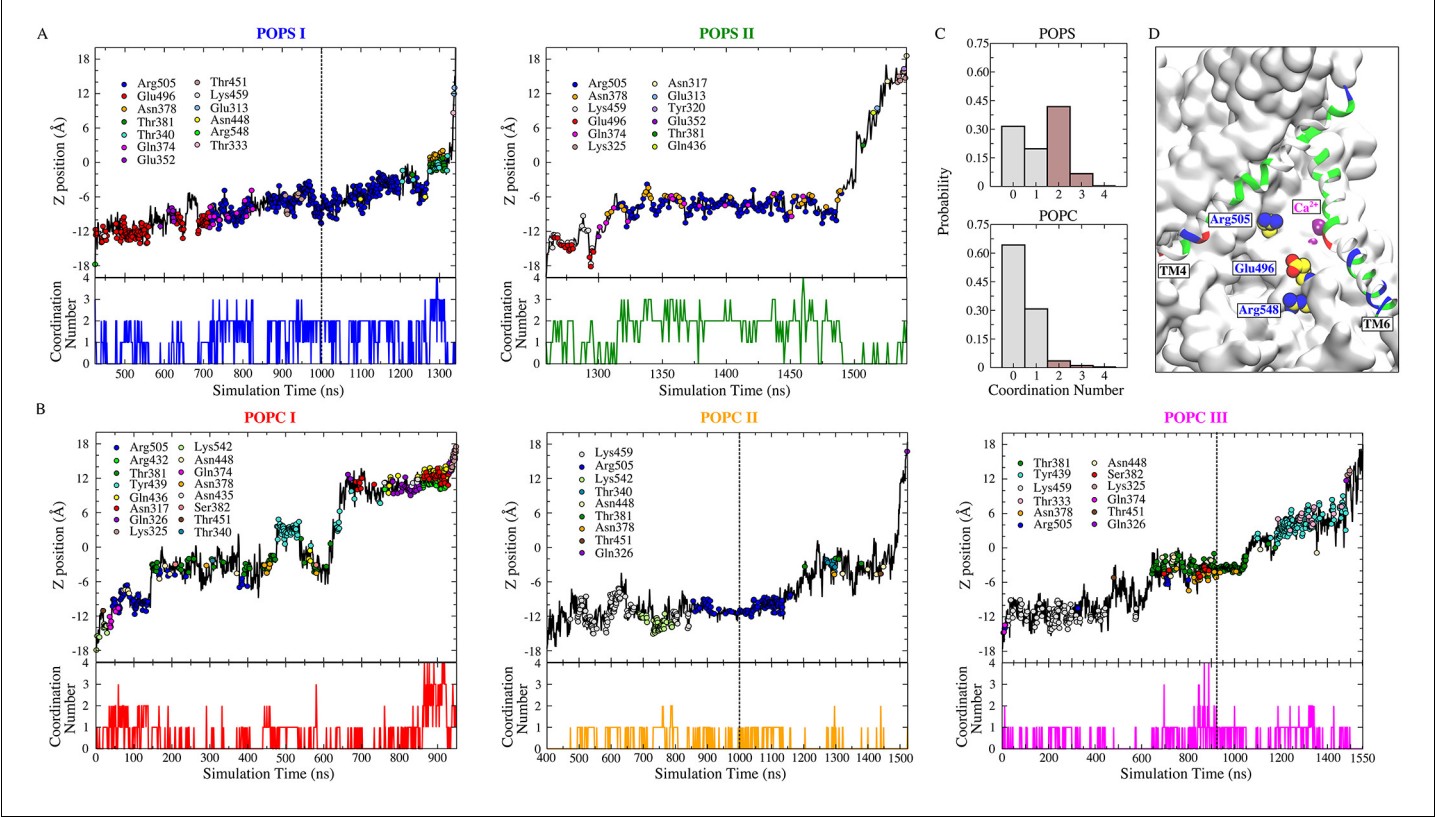

**Figure 8.** POPS is more coordinated by protein than POPC during flopping. (**A**) Full permeation of POPS from the inner leaflet to the outer leaflet was captured in subunit I (left) and II (right) during the 1700 ns simulation (1000 ns equilibrium followed by 700 ns simulation with −500 mV transmembrane potential). The dashed line indicates the time of electric field application. *Top panels:* Translocation of POPS through the aqueduct, measured as the z-position of its phosphorus atom versus time. The head group coordinating residues during lipid flopping are shown as dots color-coded to the amino acid residues in the inset. *Bottom panels:* Head group coordination number of the scrambled POPS as a function of time. (**B**) Full permeation of POPC. POPC I (red) was fully scrambled before the external electric field was applied. (**C**) Normalized probability histograms of the head group coordination number during the lipid scrambling, averaged for the scrambled POPS (top panel) and POPC (bottom panel). POPC is less coordinated by the protein compared to POPS. (**D**) Residues inside the aqueduct that strongly favor the binding of POPS over POPC are labeled (E496, R505, and R548).
DOI: https://doi.org/10.7554/eLife.28671.021

The following figure supplements are available for figure 8:

**Figure supplement 1.** Scrambling events captured during the 1700 ns simulation.
DOI: https://doi.org/10.7554/eLife.28671.022

**Figure supplement 2.** POPS binding inside the aqueduct.
DOI: https://doi.org/10.7554/eLife.28671.023

**Figure supplement 3.** Comparison of POPS and POPC head group coordination from residues E496, R505 and R548 during the 1700 ns simulation.
DOI: https://doi.org/10.7554/eLife.28671.024

**Figure supplement 4.** Lipid P-N dipole orientations for head groups inside the aqueduct (bin = 2 Å along z axis) over the simulations with different transmembrane potentials.
DOI: https://doi.org/10.7554/eLife.28671.025

## Insights into Ca²⁺ activation

In contrast to the Ca$^{2+}$-activated simulation, lipid occupancy of the aqueduct did not occur when Ca$^{2+}$ was removed. On average only 1.8 lipids were present in the aqueduct in the Ca$^{2+}$-free state compared to 3.3 the Ca$^{2+}$-liganded conformation (*Figure 9—figure supplement 1*). In the absence of Ca$^{2+}$, lipid phosphates pack near the intracellular and extracellular entrances. The observation that some lipids are located in the aqueduct even in the Ca$^{2+}$-free conformation may be consistent with the observation that nhTMEM16 and afTMEM16 reconstituted into liposomes exhibit significant phospholipid scrambling in the absence of Ca$^{2+}$ (*Malvezzi et al., 2013*; *Brunner et al., 2014*).

The mechanism of Ca$^{2+}$-dependent gating of the channel was explored by computing the RMSD values for each residue in Ca$^{2+}$-liganded and the Ca$^{2+}$-free states by least squares fitting (*Figure 9A*). Overall, N- and C- termini and extra-membrane loops exhibited the greatest differences between the two states as expected from their flexibility and B-factors in the crystal structure. Trans-membrane helices TM1, TM2, TM5, TM7, TM8, and TM9 were very similar between the two states with RMSD <2.5 Å, while TM10 exhibited a small difference (RMSD <4 Å between conformations). Only TM3, TM4, and the cytoplasmic half of TM6 showed significant changes. The greatest change is in the cytoplasmic half of TM6 (residues 451–470, RMSD = 10.6 Å) which moves away from TM7

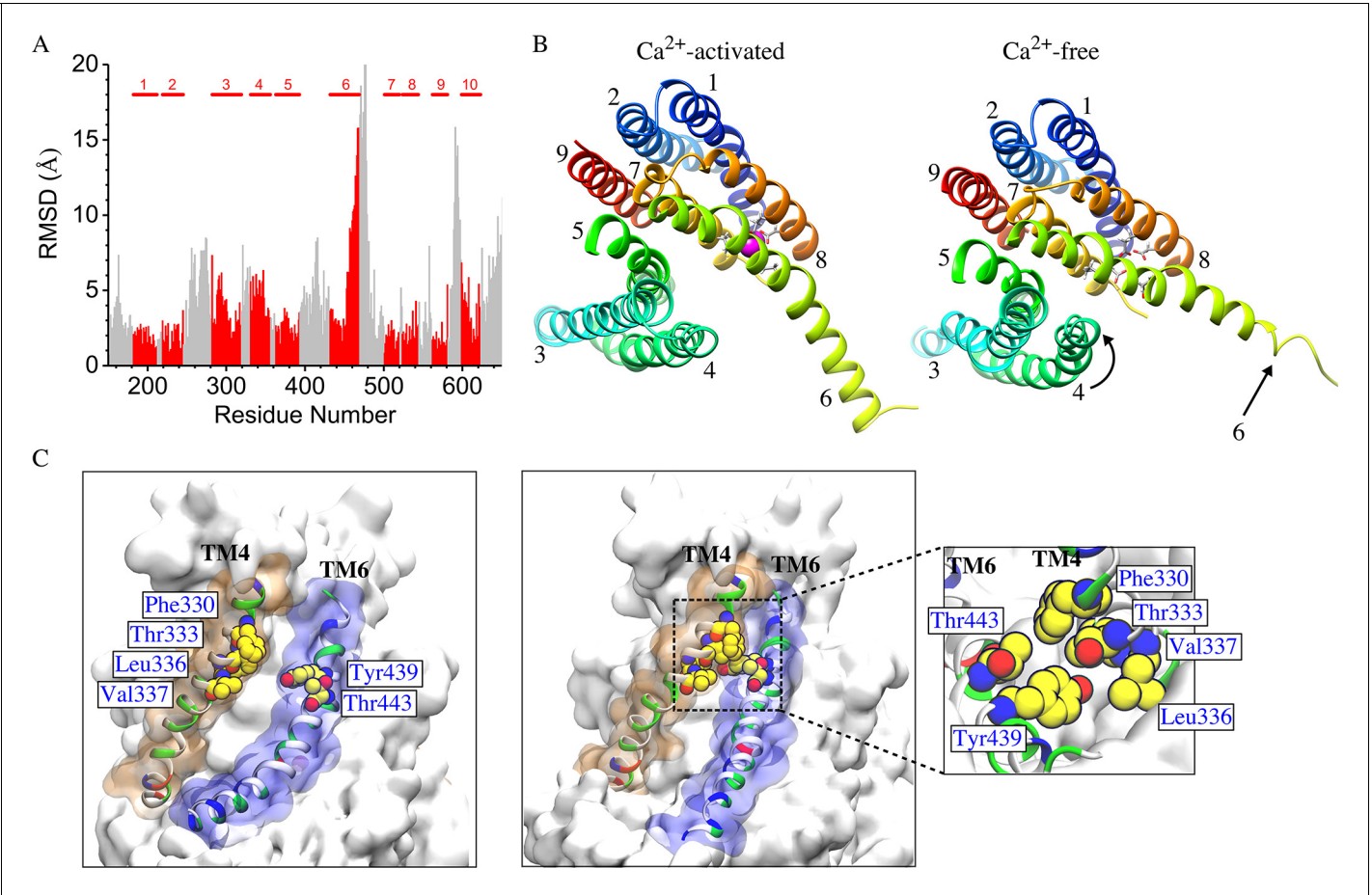

**Figure 9.** Difference between Ca$^{2+}$-free and Ca$^{2+}$-liganded nhTMEM16. (**A**) Ca$^{2+}$-free and Ca$^{2+}$-liganded nhTMEM16 were structurally aligned with UCSF Chimera and the RMSD of each residue calculated using least squares. TM domains are indicated at the top and RMSD values for TM domains are colored red. (**B**) Representative structures for Ca$^{2+}$-activated (at ~837 ns) and Ca$^{2+}$-free (at ~487 ns) nhTMEM16 from the 1 μsec simulations, respectively. The TM domains are colored and labeled. Ca$^{2+}$ ions are magenta. Ca$^{2+}$-coordinating residues are represented as sticks. (**C**) Illustration of the aqueducts in (**B**) showing the normal conformation (left panel) and the pinching near the gating region in the Ca$^{2+}$-free state (right panel). Pinching occurs when TM4 and TM6 are in close contact (sidechain-sidechain distances < 3.0 Å) near the bottleneck region. TM4 and TM6 are colored by residue type and contoured by orange and blue molecular surfaces. Residues involved in inter-helical contacts (F330, T333, L336 and V337 from TM4, Y439 and T443 from TM6) are shown as van der Waals spheres. The inset shows the top view of the pinching of the aqueduct by the hydrophobic contacts of residues from TM4 and TM6.

DOI: https://doi.org/10.7554/eLife.28671.026

The following video and figure supplements are available for figure 9:

**Figure supplement 1.** Comparison of head group distributions in the Ca$^{2+}$-activated and Ca$^{2+}$-free simulations.
DOI: https://doi.org/10.7554/eLife.28671.027

**Figure supplement 2.** Overlaid structures of Ca$^{2+}$-free (tan) and Ca$^{2+}$-activated (sky blue) nhTMEM16 as shown in*Figure 9*.
DOI: https://doi.org/10.7554/eLife.28671.028

**Figure 9—Video 1.** Movie shows a morph between the representative conformations of the Ca$^{2+}$-free and Ca$^{2+}$-activated states.
DOI: https://doi.org/10.7554/eLife.28671.029

and TM8. This movement is dramatic: Q468 moves ~15 Å relative to TM7 and TM8 which remain relatively constant. This movement probably comes about as a result of electrostatic changes in the $Ca^{2+}$ binding site when $Ca^{2+}$ is removed. In the $Ca^{2+}$-liganded conformation, two $Ca^{2+}$ ions are coordinated by E452 in TM6, D503 and E506 in TM7, and E535 and D539 in TM8. When $Ca^{2+}$ is removed these acidic amino acids repel to bring about the displacement of TM6. While this movement is very large, it seems unlikely that this movement itself is the gate, because the aqueduct is quite wide at this point. More likely, the conformational change observed in the extracellular half of TM4 (residues 325–345, RMSD = 5.1 Å) which forms a narrow constriction of the aqueduct are responsible for gating.

TM4 (and to a lesser extent TM3) become much closer to TM6 at the extracellular side of the membrane in the absence of $Ca^{2+}$ so that the aqueduct is noticeably pinched (*Figure 9B*, *Figure 9—figure supplement 2*). For example, F330 and T443 separate from 2.7 Å to 11.4 Å upon $Ca^{2+}$ binding and T333 and Y439 separate from 2.6 Å to 6.1 Å (*Figure 9C*, *Figure 9—video 1*). This close contact between residues in the outer portions of TM4 and TM6 in the $Ca^{2+}$-free conformation would be expected to sterically prevent lipids from penetrating into the aqueduct. Pinching of the aqueduct (distance <3 Å) at the level of T333 and Y439 occurs 40.3% of the time during the last 150 ns of the 1000 ns simulation in the absence of $Ca^{2+}$ and only 6.3% of the time in the presence $Ca^{2+}$ (*Figure 9C*). If this region is indeed the gate, the observation that the gate is closed less than 50% of the time in the $Ca^{2+}$-free state is consistent with reports that the purified fungal nhTMEM16s exhibit significant scrambling in the absence of $Ca^{2+}$.

The narrowing of the outer part of the aqueduct reduces the hydration of the permeation pathway. The aqueduct in the $Ca^{2+}$-liganded conformation is fully hydrated throughout its entire length, as one would expect if these structures are providing an aqueous pathway for movement of hydrophilic lipid head groups. In contrast, the aqueducts in the $Ca^{2+}$-free conformation display markedly decreased hydration especially from the center of the membrane to the outer vestibule (0 < z < 10 Å) (*Figure 10*, *Figure 10—figure supplement 1*). For example, the average number of water molecules in the aqueduct between z = 0–10 Å is decreased by >60% when $Ca^{2+}$ is removed.

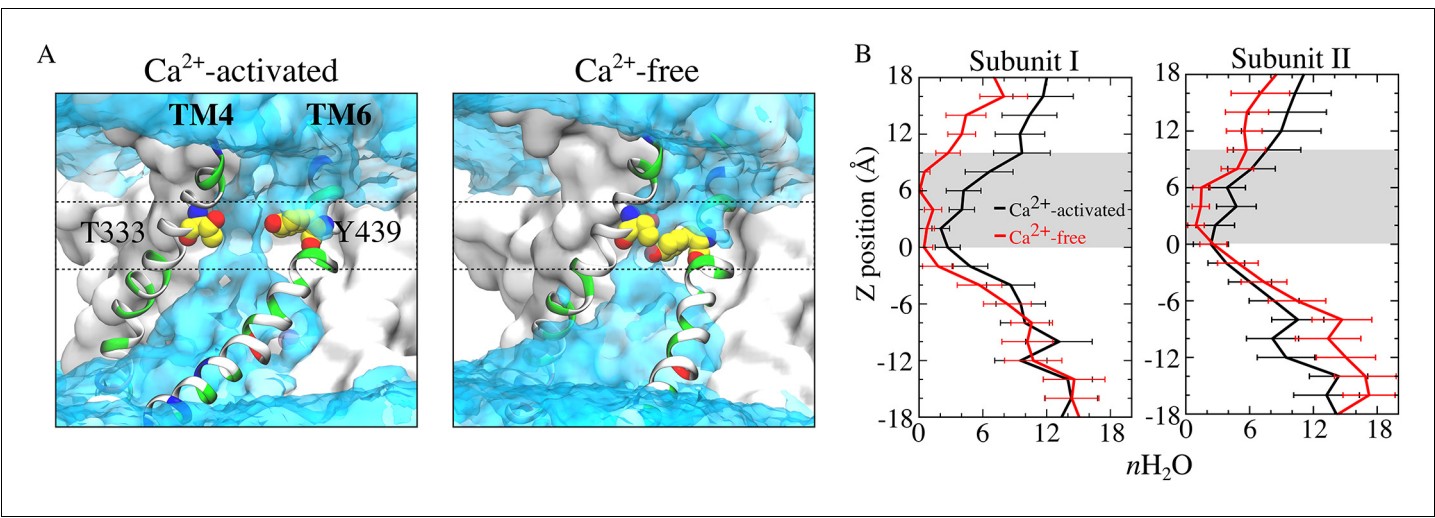

**Figure 10.** Aqueduct hydration. (**A**) Average water density of the $Ca^{2+}$-activated (left panel) and $Ca^{2+}$-free (right panel) simulations is shown as a transparent cyan surface contoured at 0.15 of bulk water density, overlaid on representative snapshots of each aqueduct. Gating residues T333 and Y439 are shown as van der Waals spheres. The gating region of the aqueduct (0 < z < 10 Å, membrane center z = 0) is highlighted by dashed lines. (**B**) Average number of water molecules during the last 150 ns of the 1000 ns of the $Ca^{2+}$-activated and $Ca^{2+}$-free simulations. The grey area represents the gating region.

DOI: https://doi.org/10.7554/eLife.28671.030

The following figure supplement is available for figure 10:

**Figure supplement 1.** Gating region of the aqueduct.
DOI: https://doi.org/10.7554/eLife.28671.031

## Discussion

The mechanism by which scramblases catalyze rapid exchange of lipids between the two leaflets of a bilayer has been a long-sought question (*Bevers and Williamson, 2016*; *Pomorski and Menon, 2016*). Unlike ion channels that transport small, spherical hydrophilic substrates, scramblases must provide a conduction mechanism that accommodates both hydrophilic head groups and extended hydrophobic tails of phospholipids at the same time. Because it is energetically costly to move the polar head group through the hydrophobic core of the membrane, scramblases must segregate the polar and non-polar regions of the amphipathic phospholipid during transport.

Even before a lipid scramblase was identified, various mechanisms were proposed for the trans-bilayer movement of lipids (*Bevers and Williamson, 2016*; *Pomorski and Menon, 2016*). One model proposed that transmembrane proteins alter lipid packing in a way that enhances the chance for a lipid to transverse the membrane (*Kol et al., 2003*; *Kol et al., 2004*). It was suggested that dynamic protein movements induced transient defects in the lipid-protein interface that favored lipids moving between leaflets. In the case of nhTMEM16, the protein clearly disrupts and thins the structure of the bilayer to lower the energy for lipids to translocate between leaflets, but this is more obviously related to hydrophobic mismatch between the transmembrane helices and the bilayer rather than to protein dynamics. Other models have suggested that scrambling could be facilitated by amphiphilic transmembrane helices on the protein surface that provide a hydrophilic surface for the passage of lipid polar head groups sequestered from the unfavorable hydrophobic environment while the lipid tails remain in the hydrophobic phase of the membrane (*Pomorski and Menon, 2006*). This appealing mechanism is strongly supported by the structure of nhTMEM16 (*Brunner et al., 2014*) and one of the main findings of the present study. This model of scrambling shares certain similarities with the proposed mechanism for ATP-dependent flippases. For example, transport of lipid-linked oligosaccharides by the ABC lipid flippase PglK involves interaction of the oligosaccharide moiety with positively charged side chains in a cavity in the protein while the lipidic polyprenyl tail remain exposed to the lipid bilayer during transport (*Perez et al., 2015*). P4-ATPases are presumed to operate by a similar mechanism (*Andersen et al., 2016*).

### Comparison to previous studies

Two previous molecular dynamic simulations of nhTMEM16 along with ours have now provided key insights into the molecular mechanisms of phospholipid scrambling. Sansom and coworkers using a coarse-grained MD simulation observed the translocation of ~15 lipid molecules between leaflets via the aqueduct during a 1-μsec simulation, but the coarse-grained simulation did not allow for characterization of individual interactions of lipid translocation and they constrain protein dynamics (*Stansfeld et al., 2015*). More recently, Bethel and Grabe (*Bethel and Grabe, 2016*) have used atomistic MD simulations combined with continuum membrane bending calculations to characterize protein-membrane interactions in TMEM16 family. Their results, like our simulations, show that nhTMEM16 produces a significant deformation of the membrane around it. When charged and hydrophilic residues of the aqueduct are neutralized computationally, they find that the energetic cost of deformation is reduced. However, the reduction is only ~20% (from ~60 to ~48 kcal/mol), suggesting that the deformation requires other parts of the protein in addition to the aqueduct.

We have concluded that there are three potential binding sites for lipid head groups in the aqueduct and have identified amino acids that interact closely with permeant phospholipid. *Bethel and Grabe (2016)* identified two sites. One site corresponds to our $S_{ext}$ site and the two amino acids they identify (E313 and R432) are also ones that we find are major interactors (*Figure 4—figure supplement 1*), interacting with lipid head groups for 32.8% and 26.1% of the total simulation time. Their cytoplasmic site, comprised of E352 and K353, is located in the wide cytoplasmic vestibule of the aqueduct at the level of lipid head groups in the inner leaflet. In our simulations, these two residues also interact strongly with lipid head groups (31.8% and 95.8% of the simulation time). In contrast, our sites $S_{cen}$ and $S_{int}$ are closer to the center of the membrane. The difference between our sites and those of Bethel and Grabe may be explained if these sites are low-affinity, low-selectivity waystations rather than binding sites in the sense of an enzymatic active site. Bethel and Grabe comment that their cytoplasmic ($S_C$) site is not an obligate stepping stone for lipids during the scrambling process because some lipids do not visit this site. The $S_C$ site is located at the very wide cytoplasmic end of the aqueduct, where lipid-protein interactions may be less constrained.

Furthermore, the interaction of lipids with the protein may depend on the lipid composition: we performed simulations in a mixture of POPC and POPS, whereas their simulations were performed in pure POPC. The amino acids T333 (TM4) and Y439 (TM6) that we identify in the gate are ones that Bethel and Grabe find form hydrogen bonds with the phosphate group of permeating lipids. We also see evidence of the dipole stacks of phospholipids noted by Bethel and Grabe, but possibly because our simulations included POPS, which do not form dipole-dipole interactions, the dipole stacks are transient. Analysis of the lipid P-N dipole orientation shows that the head group dipoles are organized inside the aqueduct (*Figure 8—figure supplement 4*): the phosphate groups precede the choline (POPC) or amino (POPS) groups in the lower half of the aqueduct (up to 4 Å) with obtuse angles respect to z axis; transition of obtuse angles to right angle was observed at ~5 Å; when the phosphate groups move above 12 Å along the aqueduct, the angles become acute and the phosphate groups thus succeed the choline or amino groups.

In an aggregate of ~3 μsec of $Ca^{2+}$-activated simulation, we have observed one complete scrambling event in the absence of voltage and four in the presence of voltage. In comparison, *Bethel and Grabe (2016)* observed four complete transitions in 4 μsec. Both of these simulations appear to be more than an order of magnitude greater than the $10^4$ lipids/sec measured experimentally for the purified fungal afTMEM16 reconstituted into liposomes by *Malvezzi et al., 2013*. The difference might be explained by the stochastic nature of individual events or by the fact that the simulations begin with a protein that is initially in an activated state that bypasses other processes that lead up to the active state in a real protein.

## Our model

Based on all the published data, we propose the following hypothesis for phospholipid scrambling by TMEM16 proteins (*Figure 11*). While the data are derived from studies exclusively on nhTMEM16, we expect that the principles apply to other TMEM16 scramblases. First, the structure of the nhTMEM16 protein introduces a bend and a thinning of the lipid bilayer to reduce the energy barrier for hydrophilic head groups to move across the membrane. Intriguingly, our analysis demonstrated that, as the simulation extended, adjacent lipids underwent significant reorientation from a simple bilayer to form a non-bilayer structure that would favor scrambling. Second, the aqueduct itself provides a hydrophilic environment for head groups to translocate between leaflets while the hydrophobic acyl chains are stabilized by interactions with hydrophobic amino acids on the lips of the aqueduct as well as acyl chains of phospholipids in the hydrophobic phase of the bilayer. Phospholipids moving through the aqueduct may be organized by virtue of interactions between head groups, but this organization may depend strongly on the species of lipid present. Third, while phospholipids do not bind strongly to specific amino acids in the aqueduct, they do cluster around specific locations that may be waystations for lipid translocation. Different phospholipids may interact with different sites depending on their charge and dimensions. One site associated with both POPC and POPS is located right above the constriction in the aqueduct. This site was identified both by us and by *Bethel and Grabe (2016)*. We propose that this constriction plays a key role in gating scrambling by $Ca^{2+}$ because this region (especially TM4) undergoes significant conformational changes in response to $Ca^{2+}$ binding. In addition, the lipid head group dipole also change its orientation in this area. The key role of T333 in gating phospholipid scrambling is supported by the T333V mutation, which greatly reduces scrambling, presumably by increasing the hydrophobicity in the gate region. This conclusion is also supported by our observation that the hydration of the aqueduct is significantly reduced in the vicinity of the constriction of the aqueduct. In the open conformation in the presence of $Ca^{2+}$, water molecules in the gate region fill up the narrowest and deepest parts of the aqueduct and provide a water-filled pathway for spontaneous diffusion of lipid head groups. This model may help explain recent observations that the fungal TMEM16 reconstituted in lipid vesicles can scramble lipids with head groups as large as ~40 Å in diameter (*Accardi, 2016*). With voltage, we observe a 3–4 Å dilation of the aqueduct which implies that the aqueduct is flexible. Clearly, our model is a first approximation and will require future simulations and experiments. In particular, simulations with more physiological lipid head group compositions and lipid tail lengths and physiological voltages are needed to fully understand the behavior of nhTMEM16 in complex bilayers. However, the model makes a number of predictions about amino acids implicated in channel gating, lipid permeation, and ion transport that can be tested experimentally.

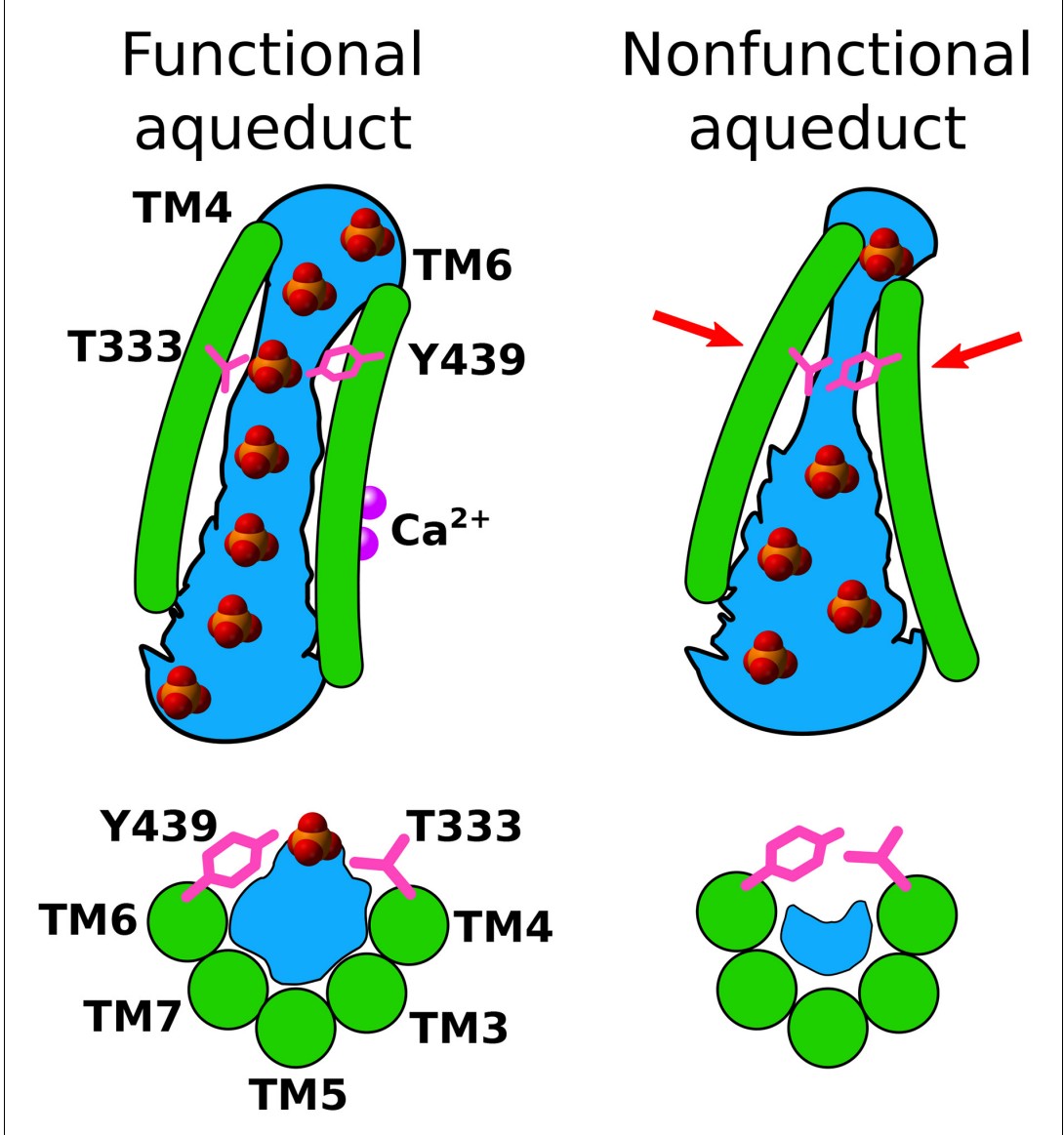

**Figure 11.** A model for $Ca^{2+}$-activated lipid scrambling. Top panels: view from the membrane. Bottom panels: view from extracellular space. Left: $Ca^{2+}$-activated. Right: $Ca^{2+}$-free. Hydration is shown in blue and $Ca^{2+}$ ions are shown as purple spheres. In the presence of $Ca^{2+}$, a hydrated pathway for lipid head groups is open, but in the absence of $Ca^{2+}$, tight contact between gating residues T333 and Y439 narrows the aqueduct, resulting in poor hydration in the gating region and prevents head groups from passing through the aqueduct.

DOI: https://doi.org/10.7554/eLife.28671.032

## Mechanisms of ion transport

Our work also provides mechanistic insights into the ion channel properties of TMEM16 proteins. Although TMEM16A and TMEM16B are clearly $Ca^{2+}$-activated $Cl^-$ channels (*Caputo et al., 2008*; *Schroeder et al., 2008*; *Yang et al., 2008*), the physiological functions of other members of the TMEM16 family have been dubious (*Duran and Hartzell, 2011*). For example, although TMEM16F certainly is a phospholipid scramblase (*Suzuki et al., 2010*), it has also been reported to be a nonselective cation channel (*Yang et al., 2012*; *Adomaviciene et al., 2013*) and various types of $Cl^-$ channel (*Almaça et al., 2009*; *Martins et al., 2011*; *Szteyn et al., 2012*; *Grubb et al., 2013*; *Shimizu et al., 2013*; *Juul et al., 2014*). The crystal structure of nhTMEM16 does not reveal an obvious alternative pathway for ion permeation through the bilayer other than the path taken by phospholipids (*Hartzell and Whitlock, 2016*) and we have proposed that the ion conductance of

TMEM16F represents a leak of ions through the aqueduct that occurs during lipid scrambling (*Yu et al., 2015*). This conclusion is based on the observation that the ionic current is relatively ion non-selective and is activated in parallel with scrambling. Initially, one stumbling block to this conclusion was the observation that the purified nhTMEM16 scramblase did not exhibit an ionic conductance (*Brunner et al., 2014*), but recently it was found that nhTMEM16 does conduct ions under certain conditions (*Lee et al., 2016*). Like the afTMEM16 fungal scramblase, the ion channel activity of nhTMEM16 is labile and depends on the composition of the lipids in which it is reconstituted. We show here that nhTMEM16 expressed in HEK cells develops weakly selective cation currents in parallel with phospholipid scrambling. The MD simulations also predict that nhTMEM16 is weakly cation-selective. Although simulations at intermediate voltages captured only $Na^+$ ion permeation, $Cl^-$ permeation was observed when high voltage was applied. In the simulations, ion conduction through the aqueous pore formed between the protein and the lipid head groups developed in parallel with phospholipid transport and became more robust as lipids became more tightly packed and organized in the aqueduct as scrambling proceeded. In this flexible pore structure, lipids play a structural role by lining the hydrophilic ion conduction pathway with their head groups. The simulation also shows that the conduction pathway is at least partly hydrated and that permeant ions are coordinated by water molecules a majority of their transit time across the membrane. These data support the hypothesis that ion transport by scramblases is a leak current associated with lipid transport. Because scrambling occurs in the absence of any ionic electrochemical driving force both in our simulations and our experiments, it seems unlikely that ion flux is necessary for phospholipid scrambling. However, the cotransport of ions along with lipids could likely modulate the lipid transport process or play some parallel physiological function.

It is likely that ion conduction by the $Cl^-$ channel TMEM16A occurs via a similar pathway to the one that conducts lipids in the TMEM16 scramblases (*Hartzell and Whitlock, 2016*; *Jeng et al., 2016*; *Lim et al., 2016*). Recently, a low-resolution cryo-EM structure of the TMEM16A $Cl^-$ channel was solved (*Paulino et al., 2017*) which showed that the TMEM16A ion-conducting pore, instead of being an open aqueduct as in nhTMEM16, is enclosed by protein along ~2/3 of its length due to a structural rearrangement that brings TM4 and TM6 closer together to enclose the pore. Otherwise the structures of TMEM16A and nhTMEM16 are very similar. The similarity in structure between TMEM16 scramblases and channels is reinforced by our finding that we can convert TMEM16A into a scramblase by the V543S (or V543T) or the K588N mutations. However, it is not obvious from the cryo-EM structure of TMEM16A how these mutations produce their effects. V543 is located near the extracellular end of TM4 and it is possible that the V543S/T mutation would 'split open' the enclosed pore of TMEM16A to convert it to an aqueduct by shifting the position of TM4. While highly speculative, this idea may not be as untenable as it might first seem. It is unclear whether the TMEM16A cryo-EM structure is an open, ion-conducting conformation; the pore is not large enough to accommodate permeant ions except possibly $Cl^-$ (*Fisher and Hartzell, 2017*). It seems likely that the new structure represents the inactivated state that TMEM16A is known to enter in the presence of high $Ca^{2+}$ concentrations used to purify the protein (*Yu et al., 2014*). In order for the channel to conduct ions as large as $I^-$ or $SCN^-$, the pore would need to dilate by at least a few Å. The V543S/T mutation might destabilize the interactions between TM4 and TM6 to open the pore to create an aqueduct. Testing this speculation will require additional simulations with a higher resolution TMEM16A structure.

The change in ionic selectivity of the V543S/T mutant from $Cl^-$-selective to non-selective as scrambling develops is challenging to explain mechanistically. Before scrambling develops, the TMEM16A V543S mutant behaves like a WT TMEM16A $Cl^-$ channel, but as scrambling develops, the channel progressively becomes less $Cl^-$ selective. This observation suggests that there is a slow conformational change in the protein that is associated with scrambling and the non-selective current. We suggest that the appearance of this non-selective current is caused by a slow conversion of the closed pore of TMEM16A into an open aqueduct. However, another possibility is that the $Cl^-$ current and the non-selective current utilize physically different conductance pathways. Experiments are underway to test these alternatives.

## Materials and methods

### Preparation of membrane-embedded scramblase

The X-ray crystallographic structure of fungal nhTMEM16 (4WIT) at 3.4 Å resolution (*Brunner et al., 2016*) was used as the starting structure for simulation. Missing loops (residues 1–18, 130–140, 465–482, 586–593, 657–659, 685–691 and 720–735) were added using SuperLooper (*Hildebrand et al., 2009*) which models loops based on templates from protein structures to fill in gaps in the protein structure. The two $Ca^{2+}$ ions bound to the two subunits were preserved for the simulation of the $Ca^{2+}$-activated system, and removed for the simulation of the $Ca^{2+}$-free system. The pKa of each ionizable residue was estimated using PROPKA (*Olsson et al., 2011*; *Rostkowski et al., 2011*) and default protonation states were assigned based on the pKa analysis. For residues inside the aqueduct that carry charges, their local environment was examined in detail. All the charged sidechains are either coordinated by polar residues in close proximity or involved in coordinating the bound ions. R505 sidechain is coordinated by N378, in addition, two energetically favorable internal water molecules were placed near R505 sidechain based on DOWSER (*Zhang and Hermans, 1996*; *Morozenko et al., 2014*) calculation. The $Ca^{2+}$ coordinating residues E452, D503, E506, E535, D539 were charged in order to coordinate the bound $Ca^{2+}$ ions. In the $Ca^{2+}$-free system, these residues were also assigned charges based on the pKa calculation. Scrutiny on the protein structure indicates that the charged state is reasonable because each of the charged sidechains is nicely coordinated by K542, Y498, N448, N531, Y188, respectively. Missing hydrogen atoms were added using PSFGEN in VMD (*Humphrey et al., 1996*). Including the four water molecules (two in each subunit) placed in the aqueduct, 109 internal water molecules were placed in energetically favorable positions within the protein using DOWSER (*Zhang and Hermans, 1996*; *Morozenko et al., 2014*).

The protein was embedded into an asymmetric lipid bilayer that was generated using the CHARMM-GUI membrane builder (*Wu et al., 2014*). The outer leaflet was composed of pure POPC (palmitoyl-oleoyl phosphatidylcholine), while the inner leaflet was a 2:1 POPC/POPS (palmitoyl-oleoyl phosphatidylserine) mixture to mimic eukaryotic plasma membranes. The membrane/protein system was then fully solvated with TIP3P water (*Jorgensen et al., 1983*) and buffered in 150 mM NaCl to keep the system neutral. The resulting $Ca^{2+}$-activated and $Ca^{2+}$-free systems consisting of ~360,000 atoms were contained in a 204 × 162 × 130 Å3 simulation box.

### Molecular dynamics (MD) simulations

MD simulations were carried out with NAMD2.9 (*Phillips et al., 2005*) using CHARMM-CMAP (*Mackerell et al., 2004*) and CHARMM36 (*Klauda et al., 2010*) parameters set to model the proteins and lipids, respectively. The particle mesh Ewald (PME) method (*Darden et al., 1993*) was used to calculate long-range electrostatic interactions every 4 fs. A smoothing function was employed for short-range non-bonded van der Waals and electrostatic interactions at a distance of 10 Å with a cutoff of 12 Å. Bonded interactions and short-range non-bonded interactions were calculated every 2 fs, and periodic boundary conditions were employed in all three dimensions. Pairs of atoms whose interactions were evaluated were searched and updated every 20 fs. A cutoff (13.5 Å) slightly longer than the non-bonded cutoff was applied to search for interacting atom pairs. Simulation systems were subjected to Langevin dynamics and the Nosé-Hoover Langevin piston method (*Nosé, 1984*; *Hoover, 1985*) to maintain constant pressure ($P$ = 1 atm) and temperature ($T$ = 310 K) (NPT ensemble).

The simulations were first optimized for 10,000 steps, followed by two 1-ns equilibrations, during which the heavy atoms and the backbone atoms from the crystalized protein structure were positionally restrained ($k$ = 2 kcal/mol/Å$^2$) to allow the added loops to relax. Then a 1000-ns unrestrained equilibrium simulation was performed.

To investigate ionic permeation by nhTMEM16, additional simulations were extended from the fully-equilibrated $Ca^{2+}$-activated simulation with different levels of voltage induced by external electric fields across the membrane. A uniform electric field $E$ was applied perpendicular to the membrane plane in the negative z direction (from extracellular to intracellular). The total potential drop across the system was calculated by $E*L_z$, where $L_z \approx 133.64$ Å is the size of the simulation box in the direction parallel to the applied field. Three simulations with $E$ = 1.12, 1.87, and 3.74 mV/Å were performed, resulting in transmembrane potentials of approximately −150, −250, −500 mV inside

relative to outside, respectively. Each electric field simulation was performed for 700 ns, during which the ionic conductivity across the membrane was determined.

## Analysis of membrane structure

The midplane of the membrane is set as the xy-plane at z = 0. The phosphorus atoms (as the center of phospholipid phosphate groups) in the outer and inner leaflets of the phospholipid bilayer were initially located at approximately z = +18 and z = −18 Å, respectively. To characterize the protein-induced membrane deformation, the locations of phospholipid phosphate groups in proximity to the protein were analyzed using the VOLMAP plugin in VMD. The shell of lipids directly affected by protein-lipid interaction was defined as lipids residing within 10 Å of the protein surface. The phosphate occupancy was averaged over the trajectory in order to calculate their density during the $Ca^{2+}$-activated simulation.

## Analysis of hydration of the aqueduct

To investigate the influence of hydration on the behavior of lipids within the aqueduct, we monitored the number of water molecules, as well as the number of lipid head group heavy atoms and fatty acid tail carbon atoms within the aqueduct during the simulations. To count water molecules in the aqueduct, a cylinder of 13 Å radius was defined for each subunit across the membrane. For each cylinder, the central axis was perpendicular to the membrane passing through the geometric center of the TM helices that line the aqueduct (TM4-TM7 and the C-terminal half of TM3). The cylinder was then divided into 2-Å-thick slabs. Water molecules inside the cylindrical region were binned into corresponding slabs based on the coordinates of the oxygen atom of each water molecule.

## Tracing lipid head groups inside the aqueduct

To trace the movement of lipid head groups along the aqueduct, the same cylindrical region defined previously was used to measure lipid head groups in the aqueduct. The z-positions of the phosphorus atoms were traced during the simulation for both subunits. If a lipid phosphorus atom entered the aqueduct from one leaflet and departed from the other end, the lipid was considered as fully transported (scrambled) by the scramblase.

## Analysis of lipid-protein interaction

To characterize the interaction of lipid head groups with potential binding sites along the aqueduct, occupancy maps of the phosphate groups over the trajectories were calculated using the VOLMAP plugin in VMD. The most visited sites within the aqueduct were identified by contouring the volumetric map of phosphate occupancy at isovalue 0.15. To determine the residues at those sites that stabilize lipid binding, the electrostatic interactions between the residues and the lipid head groups inside the aqueduct were calculated for each frame of the trajectory using a distance cutoff of 4 Å. Hydrophobic interaction between lipid inside the aqueduct and the protein was analyzed by searching the carbon-carbon contact between the protein and the lipid tails for each frame of the trajectory, using a distance cutoff of 4 Å.

## Coordination of ions and lipids

Coordination number for permeant $Na^+$ was calculated by counting the number of oxygen atoms from water molecules, protein, and lipid within 3 Å of the ion during its permeation through the aqueduct. Average coordination number was obtained by dividing the total number of coordinations during ion permeation by time. Coordination number for permeant POPC and POPS was calculated by counting the number of oxygen and nitrogen atoms from protein within 3 Å of the head group phosphorus/oxygen/nitrogen atoms during their scrambling. To investigate the lipid specificity of the scramblase, coordination ratio was calculated for each residue that participates in head group coordination for POPC or POPS over the 1700 ns simulation (1000 ns at 0 mV, followed by 700 ns at −500 mV). The coordination ratio was defined as the total number of coordinations provided by each residue for POPC or POPS over the trajectory divided by the simulation time.

## Transfections

HEK-293 cells were purchased (frozen) from ATCC in 2002 (catalogue number CRL-1573) and were authenticated using STR profiling. Whilst HEK cells are included in the list of commonly misidentified cell lines maintained by the International Cell Line Authentication Committee, they are a standard cell line that is widely used for heterologous expression of ion channels because it expresses very few endogenous ion channels. For our purposes, this cell line is useful because it does not express $Ca^{2+}$-activated $Cl^-$ currents. Tests for mycoplasma were negative. Cells were maintained in modified DMEM supplemented with 10% FBS, 100 U/ml penicillin G and 100 µg/ml streptomycin) and transiently transfected with mTMEM16A (Uniprot Q8BHY3), mTMEM16F (Uniprot Q6P9J9), or nhTMEM16 (Uniprot C7Z7K1). TMEM16A and TMEM16F were tagged on the C-terminus with EGFP. Codon-optimized nhTMEM16 was designed and synthesized by DNA2.0 (Newark, CA) to optimize expression and was tagged on the N-terminus with 3X-FLAG by subcloning into the 3X-FLAG-myc-cmv26 plasmid (Sigma-Aldrich, St. Louis, MO). PCR-based mutagenesis was used to generate single amino acid mutations. All mutations were verified by sequencing.

## Phospholipid scrambling assay

Scrambling was assessed in single cells by elevating intracellular $[Ca^{2+}]$ via the patch pipet solution during patch clamp recording as described previously (*Yu et al., 2015*). Binding of Annexin-V-Alexa-Fluor-568 to patch-clamped cells during voltage-clamp recording was imaged with a wide-field Zeiss Axiovert 100 microscope using a 40X NA 0.6 LD-Acroplan objective. Images were acquired with an Orca-FLASH 4.0 digital CMOS camera (C11440, Hamamatsu) controlled by Metamorph 7.8 software (Molecular Devices). Annexin-V-AlexaFluor-568 was added to the normal extracellular solution before patch clamping the cell. After whole-cell recording was established with an intracellular solution containing 200 µM free $Ca^{2+}$ the accumulation of Annexin-V on the plasma membrane was imaged at 1 min intervals synchronously with voltage clamp recording.

## Electrophysiology

Single transfected HEK-293 cells were identified by fluorescence on a Zeiss Axiovert microscope and voltage-clamped using conventional whole-cell patch-clamp techniques with an EPC-7 amplifier (HEKA, Germany). Fire-polished borosilicate patch pipettes were 3–5 MOhm. Experiments were conducted at ambient temperature (24–26°C). Liquid junction potentials calculated using pClamp were predicted to be <2 mV. Zero $Ca^{2+}$ intracellular solution contained (mM): 146 CsCl, 2 $MgCl_2$, 5 EGTA, 10 sucrose, 10 HEPES pH 7.3 adjusted with NMDG; 20 µM $Ca^{2+}$ intracellular solution contained 5 $Ca^{2+}$-EGTA instead of 5 EGTA; 0.2 mM $Ca^{2+}$ intracellular had 0.2 $CaCl_2$ added to the 20 µM $Ca^{2+}$ solution; standard extracellular solution contained 140 NaCl, 5 KCl, 2 $CaCl_2$, 1 $MgCl_2$, 15 glucose, 10 HEPES pH 7.4. For determining ionic selectivity, the standard extracellular solution contained various concentrations of NaCl, CsCl, or NMDG-Cl as indicated and the internal solution was 150 NaCl (or CsCl), 1 $MgCl_2$, 5 Ca-EGTA, 0.2 $CaCl_2$, 1 HEPES pH 7.4. The osmolarity of each solution was adjusted to 300 mOsm by addition of mannitol. Relative permeabilities of cations relative to $Cl^-$ were determined by measuring the changes in zero-current $E_{rev}$ using the Goldman-Hodgkin-Katz equation when the concentration of extracellular ions was changed ('dilution potential' method) as previously described (*Yu et al., 2012*)

$$\Delta E_{rev} = 25.7 \ln \left[ (X_0 + C1_i * P_{Cl}/P_{Na})/(X_i + C1_0 * P_{Cl}/P_{Na}) \right]$$

where $p$ is permeability, $X_0$ is the extracellular cation concentration and $X_i$ is the intracellular cation concentration and $\Delta E_{rev}$ is the difference between $E_{rev}$ with the test solution $XCl$ and that observed with symmetrical solutions.

## Immunostaining

HEK-293 cells transiently expressing 3X-FLAG-nhTMEM16 were plated on glass coverslips, fixed in phosphate-buffered 4% paraformaldehyde for 10 min at room temperature, permeabilized with 1.5% Triton X-100, and stained with anti-FLAG M2 antibody (Sigma-Aldrich) for two hours at room temperature. Coverslips were washed and stained with goat-anti-mouse-Alexa-488 or goat-anti-mouse-Alexa-568 secondary (1:1000, Molecular Probes).

## Acknowledgement

This research is supported by grants from the National Institutes of Health R01-GM086749, R01-GM123455, U54-GM087519, and P41-GM104601 to ET, and R01-EY0114852 and R01-AR067786 to HCH, and a grant from the Muscular Dystrophy Foundation to HCH. Simulations in this study have been performed using allocations at National Science Foundation Supercomputing Centers (XSEDE grant number MCA06N060), and at the NCSA Blue Waters.

## Additional information

### Funding

| Funder | Grant reference number | Author |
|---|---|---|
| National Institutes of Health | R01-GM086749 | Emad Tajkhorshid |
| National Institutes of Health | U54- GM087519 | Emad Tajkhorshid |
| National Institutes of Health | P41-GM104601 | Emad Tajkhorshid |
| National Institutes of Health | R01-EY0114852 | H Criss Hartzell |
| National Institutes of Health | R01-AR067786 | H Criss Hartzell |
| National Science Foundation | MCA06N060 | Emad Tajkhorshid |
| National Institutes of Health | R01-GM123455 | Emad Tajkhorshid |
| Muscular Dystrophy Association | Research Grant | H Criss Hartzell |
| National Centre for Supercomputing Applications | Blue Waters | Emad Tajkhorshid |

The funders had no role in study design, data collection and interpretation, or the decision to submit the work for publication.

### Author contributions

Tao Jiang, Conceptualization, Data curation, Formal analysis, Investigation, Visualization, Methodology, Writing—original draft, Writing—review and editing; Kuai Yu, Data curation, Formal analysis, Investigation, Methodology; H Criss Hartzell, Conceptualization, Supervision, Funding acquisition, Investigation, Writing—original draft, Project administration, Writing—review and editing; Emad Tajkhorshid, Conceptualization, Resources, Supervision, Funding acquisition, Investigation, Writing—original draft, Project administration, Writing—review and editing

### Author ORCIDs

Tao Jiang http://orcid.org/0000-0002-4359-1475
H Criss Hartzell https://orcid.org/0000-0002-3393-1528
Emad Tajkhorshid http://orcid.org/0000-0001-8434-1010

### Decision letter and Author response

Decision letter https://doi.org/10.7554/eLife.28671.034
Author response https://doi.org/10.7554/eLife.28671.035

## Additional files

### Supplementary files

• Transparent reporting form
DOI: https://doi.org/10.7554/eLife.28671.033

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
