## [Decision Letter]

Thank you for submitting your article "Lipids and ions traverse the membrane by the same physical pathway in the nhTMEM16 scramblase" for consideration by *eLife*. Your article has been favorably evaluated by Gary Westbrook (Senior Editor) and three reviewers, one of whom, Nir Ben-Tal (Reviewer #1), is a member of our Board of Reviewing Editors. The following individual involved in review of your submission has agreed to reveal their identity: Raimund Dutzler (Reviewer #2). The reviewers have discussed the reviews with one another and the Reviewing Editor has drafted this decision to help you prepare a revised submission.

Summary:

Based on our discussion of your manuscript we agree that the study provides an interesting molecular perspective of the mechanisms of lipid and ion transport in the TMEM16 family. While there have been both experimental and simulation studies performed on this system before, the combined approach is a strong point of the work. The manuscript provides a detailed and stimulating view on different aspects of the function and regulation of a Ca^2+^-activated lipid scramblase. In general, we feel that it captured the essence of how a lipid scramblase catalyzes the bidirectional diffusion of lipids between both leaflets of the bilayer. Other proposals made in the manuscript, such as the activation of the protein by Ca^2+^, the permeation of ions and the conversion of the channel TMEM16A into a scramblase by point mutations are at this stage speculative, but they put forward testable hypotheses that will stimulate further studies. However, there are also a number of potential pitfalls both with simulations and experiments (not to mention models) that we feel should be addressed better.

We do not think that new simulations or experiments are necessary, but the reviewers and editors would like to see a much more critical distinction between solid and speculative aspects taking potential ambiguities in the results into account, as well as addressing the issues below e.g. with new analysis of existing data.

Essential revisions:

General:

The manuscript is rather long and a bit tedious to read. Please edit to make more concise so it is easier to read for non nhTMEM16 scramblase fascinados.

The results on point mutants of TMEM16A that confer scrambling activity to the protein and that changes the apparent selectivity of currents over time from anion selective to non-selective is puzzling, particularly in light of a recent structural investigation that showed a different organization of the ion conduction path compared to scramblases. Please address this in the Discussion.

The region the authors refer to as aqueduct is equivalent to the region that in the manuscript describing the structure was termed 'subunit cavity'. Please clarify in the manuscript.

MD simulations:

Some of the MD simulations rely on very limited data (sometimes only single events) from statistical mechanics, and as the authors cite this system has previously been studied with simulations by other groups. The authors should clearly stress where they agree with previous simulations, what results are new/different, and analyze/comment on the statistical certainty of new results. In light of the high complexity of the system and the extended length of trajectories, it would also be interesting for the reader to know more about the convergence of the system during the simulation and the structural relationships between the observed conformations in the simulated systems compared to the initial crystal structure. This might be particularly relevant for simulations with applied external potential.

When combined with experimental data, an obvious critical test is to perform additional simulations that include the mutations identified to switch the activity on/off, and show that they have the expected effects. Although we do not require the authors to perform such additional studies, it would be valuable to outline specific predictions about other experimental/simulation results that are expected if this model is correct. Is it possible to confirm other data (such as ion conductance) with experimental results, e.g., by extrapolating the applied voltages to physiologically relevant potentials?

Although it is common to use higher potentials in simulations, the authors should justify their choice of going up to 500mV, explain to the reader that it is significantly higher than physiological potentials. Are there other potential problems than distortion of the structure?

The flipping rates observed in the simulations appear to be orders of magnitude higher than the experiments by Malvezzi et al. [Nat Commun 4, 2367] that suggests rates of ~10,000 lipids per minute. This casts some doubt on the qualitative accuracy of simulations; The authors should explain or acknowledge this discrepancy. Are there other features where the simulations are in better quantitative agreement with experimental kinetics?

Although phospholipids are the most important substrates, there are indications e.g. ceramides (without the phosphate group) can be flipped by these proteins [Suzuki et al., J Biol Chem 288, 13305]. Given that the model reported here appears to rely heavily on electrostatic interactions, how would it explain the activity for these other types of lipids?

One of the main arguments of the work is that the ion conductance pathway should be identical to the scramblase pathway. However, at least for TMEM16A there is now a recent (low-resolution) structure available [Paulino, *eLife* 2017;6:e26232] where the structure is found to be rearranged compared to nhTMEM16, which creates a mostly occluded ion permeation pathway. There is no question the authors' V543S mutant is interesting, but given the complex interplay with changes in ion selectivity after phospholipid scrambling is initiated, would this not rather suggest potential structural transitions that open/close the ion pore vs. creating the surface necessary for scrambling? Obviously, this structure was not available when the authors performed their work, but even with limited resolution it appears to be in partial conflict with the present statement that there is no difference in ion permeation pathways from that taken by phospholipids. For the readers' sake, the model should be compared to these results and the differences discussed.

What is the connection between the absence of bound Ca^2+^ and the conformational change closing the lipid pathway?

Why did the authors observe permeation of Na^+^ but not Cl^-^ ions? This is in conflict with the poor current selectivity observed in experiment.

Cellular assays:

It would be interesting to learn more about the technical limitations of the cellular assay used in this study. The authors apply Ca^2+^ present in their patch pipette to the cytoplasm after establishing a whole-cell patch. It is expected that the equilibration of the Ca^2+^ concentration with cytosol would proceed within a minute. The cellular response indicating lipid scrambling was obtained after 6 minutes and the measurement lasted 14 minutes. It would be interesting to learn more about any non-specific cellular response to high (200 µM) Ca^2+^ from mock-transfected cells. How long would it take before cellular artifacts become apparent and how would these look like with respect to AnnexinV-binding and patch-clamp experiments.

The accumulation of phosphatidyl-serine on the outside of the plasma membrane, which is a prerequisite for the detection by AnnexinV, takes time, but what is the underlying cause for the delay in the activation of currents? The anticipation is instantaneous activation of the scramblase, and currents.

Have the authors considered an increase of the intracellular Ca^2+^ concentration by Ca^2+^ ionophores instead of a patch pipette? If so, did these experiments show a similar result with respect to scrambling?

Why was the codon optimized, N-terminally-tagged construct of nhTMEM16 used for experiments in Figure 5 not used for experiments shown in Figure 4?

In vitro assays have previously indicated a basal activity of nhTMEM16 in the absence of Ca^2+^. Do the authors have evidence for such activity in their assays?

The results describing the behavior of TMEM16A mutants are puzzling. Why do 20% of cells expressing WT show scrambling? What is the evidence that the currents measured from cells transfected with the TMEM16A mutant V543S after 15 min are still mediated by this protein? How would WT currents measured under the same conditions look like?

Why does the mutant K588N but not K588Q confer scrambling activity to TMEM16A given that, according to a recent structure at low resolution, the residue might not be exposed to the membrane?

---

## [Author Response]

Essential revisions:General:The manuscript is rather long and a bit tedious to read. Please edit to make more concise so it is easier to read for non nhTMEM16 scramblase fascinados.

The manuscript has been completely re-written. Although we feel that the manuscript is now more concise and tightly written, its length has not changed significantly because substantive responses to the reviewers’ comments compensated for the words that were deleted.

The results on point mutants of TMEM16A that confer scrambling activity to the protein and that changes the apparent selectivity of currents over time from anion selective to non-selective is puzzling, particularly in light of a recent structural investigation that showed a different organization of the ion conduction path compared to scramblases. Please address this in the Discussion.

We agree that this observation is hard to explain structurally. However, the observation is reproducible and consistent. Although we have additional data on this issue, we have chosen not to add it to this paper because it is somewhat tangential to the main point of this paper and it would add significantly to the length of the present manuscript. We have added a section in Discussion about this:

“It is likely that ion conduction by TMEM16A occurs via a similar pathway to the one that conducts lipids in the TMEM16 scramblases (Lim et al., 2016; Jeng et al., 2016; Hartzell and Whitlock, 2016). […] However, another possibility is that the Cl^–^ current and the non-selective current utilize physically different conductance pathways. Experiments are underway to test these alternatives.”

The region the authors refer to as aqueduct is equivalent to the region that in the manuscript describing the structure was termed 'subunit cavity'. Please clarify in the manuscript.

We agree. The terminology in Introduction has been changed to “subunit cavity”. Towards the end of Introduction we introduce the term aqueduct and explain its usage:

“Because this surface cavity is hydrophilic and involved in substrate transport, we call it the "aqueduct"”.

MD simulations:Some of the MD simulations rely on very limited data (sometimes only single events) from statistical mechanics, and as the authors cite this system has previously been studied with simulations by other groups. The authors should clearly stress where they agree with previous simulations, what results are new/different, and analyze/comment on the statistical certainty of new results. In light of the high complexity of the system and the extended length of trajectories, it would also be interesting for the reader to know more about the convergence of the system during the simulation and the structural relationships between the observed conformations in the simulated systems compared to the initial crystal structure. This might be particularly relevant for simulations with applied external potential.

Throughout the manuscript, we have added quantification of the data and have clarified the number of observations. For the single events (lipid scrambling under equilibrium condition, and ion conduction under low transmembrane potentials) observed in the MD simulations, multiple occurrences were captured after transmembrane potential was added (for lipid scrambling) or increased (for ion conduction). Those repeated events were analyzed statistically, in addition to the individual analysis of the single events (Figure 7—figure supplement 1, Figure 7—figure supplement 2, Figure 8).

We have also added a section in Discussion entitled “Comparison to previous studies” comparing our results to those of other groups. The convergence of the conformations and the difference from the crystal structure are also discussed (subsection “The lipid translocation pathway”, first paragraph). For structural comparison between the simulated systems (with applied external potential) and the initial crystal structure, figures (Figure 7—figure supplement 4 and Figure 7—figure supplement 5) and analysis have been added (subsection “The aqueduct also conducts ions in simulations”, last paragraph). please also see the response to the 3^rd^ question in “MD simulations”.

When combined with experimental data, an obvious critical test is to perform additional simulations that include the mutations identified to switch the activity on/off, and show that they have the expected effects. Although we do not require the authors to perform such additional studies, it would be valuable to outline specific predictions about other experimental/simulation results that are expected if this model is correct. Is it possible to confirm other data (such as ion conductance) with experimental results, e.g., by extrapolating the applied voltages to physiologically relevant potentials?

While we have not explicitly proposed predictions of our models, we hope that the re-writing of the manuscript has addressed these concerns. (Please also see answer to “General Question” 2, above.)

Currently, there are no experimental data for ion conductance in nhTMEM16. The ion conductance for afTMEM16 is ~300 pS (Malvezzi et al. (2013)). Based on the simulation performed under -500 mV membrane potential, the ion conductance (24 conduction events) over the whole trajectory (700ns) is ~11 pS, and ~32 pS for the dilated state (Figure 7). In addition, we performed a 90-ns simulation at -1V, in which 38 Na^+^ influx and 8 Cl^-^ efflux were captured. Based on this short simulation, the ion conductance is ~82 pS over the whole trajectory, and ~235 pS during the highly conductive phase (t=60-90 ns, Figure 7—figure supplement 3).

Although it is common to use higher potentials in simulations, the authors should justify their choice of going up to 500mV, explain to the reader that it is significantly higher than physiological potentials. Are there other potential problems than distortion of the structure?

This has been addressed:

“These large non-physiological voltages were used to increase the frequency of permeation events during the 700 ns simulations by increasing the driving force for ion flux. […] At -150 mV and -250 mV, the TM4-TM6 COM distances are very similar to those in the crystal structure and simulation at 0 mV (Figure 7—figure supplement 5); the average COM distances in simulation at -500 mV is slightly larger, due to the transient dilation phase; the “normal state”, which takes the major length of the simulation, is very similar to that in the crystal structure (Figure 7—figure supplement 4(A)).”

The flipping rates observed in the simulations appear to be orders of magnitude higher than the experiments by Malvezzi et al. [Nat Commun 4, 2367] that suggests rates of ~10,000 lipids per minute. This casts some doubt on the qualitative accuracy of simulations; The authors should explain or acknowledge this discrepancy. Are there other features where the simulations are in better quantitative agreement with experimental kinetics?

Paragraph added to Discussion:

“In an aggregate of ~3 µsec of Ca^2+^-activated simulation, we have observed 1 complete scrambling event in the absence of voltage and 4 in the presence of voltage. […] The difference might be explained by the stochastic nature of individual events or by the fact that the simulations begin with a protein that is initially in an activated state that bypasses other processes that lead up to the active state in a real protein.”

Although phospholipids are the most important substrates, there are indications e.g. ceramides (without the phosphate group) can be flipped by these proteins [Suzuki et al., J Biol Chem 288, 13305]. Given that the model reported here appears to rely heavily on electrostatic interactions, how would it explain the activity for these other types of lipids?

The reviewer may have misunderstood our model. The confusion may have originated from our use of the phosphate group for the analysis and for the figures when trying to describe the motion of the lipids. Our model does not depend heavily on what is described in the comment as electrostatic interactions (net charges). Indeed, most of the residues that interact with permeant POPC head groups are not charged. In the case of POPS, however, there is more interaction with basic amino acids. We have added a section in the Discussion that clearly outlines the essential features of our model. Any polar groups, such as hydroxyl and amino groups present in ceramides, can use the aqueduct to permeate.

**“**Our model. Based on all the published data, we propose the following hypothesis for phospholipid scrambling by TMEM16 proteins. […] Different phospholipids may interact with different sites depending on their charge and dimensions.**”**

• One of the main arguments of the work is that the ion conductance pathway should be identical to the scramblase pathway. However, at least for TMEM16A there is now a recent (low-resolution) structure available [Paulino, eLife 2017;6:e26232] where the structure is found to be rearranged compared to nhTMEM16, which creates a mostly occluded ion permeation pathway. There is no question the authors' V543S mutant is interesting, but given the complex interplay with changes in ion selectivity after phospholipid scrambling is initiated, would this not rather suggest potential structural transitions that open/close the ion pore vs. creating the surface necessary for scrambling? Obviously, this structure was not available when the authors performed their work, but even with limited resolution it appears to be in partial conflict with the present statement that there is no difference in ion permeation pathways from that taken by phospholipids. For the readers' sake, the model should be compared to these results and the differences discussed.

A section has been added to Discussion addressing these issues (subsection “Mechanisms of Ion Transport”, last two paragraphs). (Please also see answer to “General Question” 2, above.)

What is the connection between the absence of bound Ca^2+^ and the conformational change closing the lipid pathway?

We observe that removal of Ca^2+^ results in a large movement of the cytoplasmic portion of TM6 into the aqueduct towards TM4. However, we do not think that this is the “gate”. Rather, a smaller conformational change in the outer region of TM4 near T333 pinches the aqueduct closed. We have expanded our analysis of the conformational changes associated with Ca^2+^ binding and have added a discussion of these findings in the section "Our model" in Discussion.

“The mechanism of Ca^2+^-dependent gating of the channel was explored by computing the RMSD values for each residue in Ca^2+^-activated and the Ca^2+^-free states by least squares fitting (Figure 9)). […] If this region is indeed the gate, the observation that the gate is closed less than 50% of the time in the Ca^2+^-free state is consistent with reports that the purified fungal nhTMEM16s exhibit significant scrambling in the absence of Ca^2+^.”

“We propose that this constriction plays a key role in gating scrambling by Ca^2+^ because this region (especially TM4) undergoes significant conformational changes in response to Ca^2+^ binding. […] However, the model makes a number of predictions about amino acids implicated in channel gating, lipid permeation, and ion transport that can be tested experimentally.”

Why did the authors observe permeation of Na^+^ but not Cl^-^ ions? This is in conflict with the poor current selectivity observed in experiment.

We find nhTMEM16 is weakly cation-selective (P_Na_/P_Cl_ >2), so it is not surprising that we see more Na^+^ than Cl^-^. At higher voltages (-1 V), we did observe Cl^-^ permeation:

**“**The probability of Na^+^ permeation increased with voltage. At -500 mV there were 24 Na^+^ influx events over 700 ns (Figure 7), Figure 7—figure supplement 2, Figure 7—video 1) and at -1000 mV there were 38 Na^+^ influx and 8 Cl^-^ efflux events over 90 ns of simulation (Figure 7—figure supplement 3)).**”**

Cellular assays:It would be interesting to learn more about the technical limitations of the cellular assay used in this study. The authors apply Ca^2+^ present in their patch pipette to the cytoplasm after establishing a whole-cell patch. It is expected that the equilibration of the Ca^2+^ concentration with cytosol would proceed within a minute. The cellular response indicating lipid scrambling was obtained after 6 minutes and the measurement lasted 14 minutes. It would be interesting to learn more about any non-specific cellular response to high (200 µM) Ca^2+^ from mock-transfected cells. How long would it take before cellular artifacts become apparent and how would these look like with respect to AnnexinV-binding and patch-clamp experiments.

The limitations of the assay have been discussed to some degree previously in Yu et al., 2015. The reason for the lag is puzzling, but a consistent feature that is discussed in our response to the next question. We do not see scrambling or currents activated in untransfected cells and this has also been shown previously in Yu et al.

The accumulation of phosphatidyl-serine on the outside of the plasma membrane, which is a prerequisite for the detection by AnnexinV, takes time, but what is the underlying cause for the delay in the activation of currents? The anticipation is instantaneous activation of the scramblase, and currents.

This is a consistent observation and was also reported in our paper on TMEM16F (Yu et al. *eLife*, 2015). TMEM16F currents are activated very quickly in excised patches (Yang et al., 2012) and we have confirmed this. However, in whole cell recording the currents and scrambling activate slowly even though global Ca^2+^ equilibrates quickly. Under similar conditions, TMEM16A activates almost instantly, yet TMEM16F takes minutes to activate. We believe that there is an endogenous inhibitor that must be removed for the current – and scrambling – activate. We are working on this.

Have the authors considered an increase of the intracellular Ca^2+^ concentration by Ca^2+^ ionophores instead of a patch pipette? If so, did these experiments show a similar result with respect to scrambling?

We find that scrambling induced by Ca^2+^ in the patch pipet is more reproducible than elevation of Ca^2+^ by ionomycin. However, initial experiments were performed on cell monolayers expressing the V543S/T mutations stimulated with ionomycin and similar results were obtained.

Why was the codon optimized, N-terminally-tagged construct of nhTMEM16 used for experiments in Figure 5 not used for experiments shown in Figure 4?

The same construct was used in all experiments. The description of the construct was moved to the section describing Figure 4 (subsection “Pinpointing residues controlling scrambling”, second paragraph).

In vitro assays have previously indicated a basal activity of nhTMEM16 in the absence of Ca^2+^. Do the authors have evidence for such activity in their assays?

In the cellular assays, ATP-dependent flippases and floppases would presumably oppose and cancel out any slow scrambling. In any case, we have not studied this experimentally. In the simulations, we comment in several places that our data may be consistent with a certain level of Ca^2+^-independent scrambling:

“On average only 1.8 lipids were present in the aqueduct in the Ca^2+^-free state compared to 3.3 the Ca^2+^-activated conformation (Figure 9—figure supplement 1). […] The observation that some lipids are located in the aqueduct even in the Ca^2+^-free conformation may be consistent with the observation that nhTMEM16 and afTMEM16 reconstituted into liposomes exhibit significant phospholipid scrambling in the absence of Ca^2+^ (Malvezzi et al., 2013; Brunner et al., 2014).”

“This close contact between residues in the outer portions of TM4 and TM6 in the Ca^2+^-free conformation would be expected to sterically prevent lipids from penetrating into the aqueduct. […] If this region is indeed the gate, the observation that the gate is closed less than 50% of the time in the Ca^2+^-free state is consistent with reports that the purified fungal nhTMEM16s exhibit significant scrambling in the absence of Ca^2+^.”

The results describing the behavior of TMEM16A mutants are puzzling. Why do 20% of cells expressing WT show scrambling? What is the evidence that the currents measured from cells transfected with the TMEM16A mutant V543S after 15 min are still mediated by this protein? How would WT currents measured under the same conditions look like?

The 20% scrambling of ANO1-expressing cells is a consistent observation. We do not know the explanation for this, but it may be related to endogenous scramblases (both TMEM16 and XKR8 -mediated). We have previously published (Yu et al. 2015) that there is neither scrambling nor currents develop in untransfected cells.

Why does the mutant K588N but not K588Q confer scrambling activity to TMEM16A given that, according to a recent structure at low resolution, the residue might not be exposed to the membrane?

We do not know, but one possible explanation is that the sidechain of N is considerably shorter (3.7 Å) than Q (4.9 Å) or K (6.2 Å). While K and Q might be capable of interacting with other residues (for example E555 in TM4) to stabilize a pore-like structure, N might be incapable of this because of its smaller size. A statement has been added:

“One possible explanation is that the side chain of N is considerably shorter (3.7 Å) than Q (4.9 Å) or K (6.2 Å). Q and K side chains may be long enough to interact with nearby amino acids to stabilize the Cl^-^ channel conformation of TMEM16A, while N may favor a scramblase conformation because it cannot make these interactions.”